# Mesoporous carbon spheres with programmable interiors as efficient nanoreactors for $H_2O_2$ electrosynthesis

Qiang Tian [1,2], Lingyan Jing [2,3] ✉, Hongnan Du[4], Yunchao Yin[1,2], Xiaolei Cheng[1], Jiaxin Xu[1], Junyu Chen[1], Zhuoxin Liu [1], Jiayu Wan[5], Jian Liu[4] & Jinlong Yang [1] ✉

The nanoreactor holds great promise as it emulates the natural processes of living organisms to facilitate chemical reactions, offering immense potential in catalytic energy conversion owing to its unique structural functionality. Here, we propose the utilization of precisely engineered carbon spheres as building blocks, integrating micromechanics and controllable synthesis to explore their catalytic functionalities in two-electron oxygen reduction reactions. After conducting rigorous experiments and simulations, we present compelling evidence for the enhanced mass transfer and microenvironment modulation effects offered by these mesoporous hollow carbon spheres, particularly when possessing a suitably sized hollow architecture. Impressively, the pivotal achievement lies in the successful screening of a potent, selective, and durable two-electron oxygen reduction reaction catalyst for the direct synthesis of medical-grade hydrogen peroxide disinfectant. Serving as an exemplary demonstration of nanoreactor engineering in catalyst screening, this work highlights the immense potential of various well-designed carbon-based nanoreactors in extensive applications.

Natural reactors, such as cells and organelles, provide a micrometer-scale hollow space and a suitable internal microenvironment for biochemical processes[1]. Their well-organized interiors enable reactants to be oriented and induced in a regulated way to perform specific functions[2,3]. Inspired by these cellular structures, various artificial nanoreactors have been designed and synthesized, making them a popular catalytic material for extensive applications[4–6]. Among them, mesoporous hollow nanoreactors (MHNs) are a novel type of catalytic material. Their unique internal hollow space and mesoporous structure can accommodate complex catalytic processes[7–9]. In particular, the hollow space in MHNs provides a confined internal microenvironment, allowing for modulation of molecular catalytic behavior[10–12]. Additionally, the mesoporous channels could provide the opportunity to control molecular diffusion, adsorption, and surface reactions for tunable catalytic reaction pathway[13–15].

The development of MHNs presents challenges in understanding reaction processes at micro/nano interface[16]. Nevertheless, MHNs possess the potential to modulate the reaction process, and its nanoscale structural parameters can be tailored to regulate mass transfer and microenvironment, thereby enabling the tailoring of catalytic reaction pathway[17–19]. Yet, the effect of structural parameters on overall catalytic kinetics is not fully understood due to the limitations in model material synthesis, which presents a significant

[1]Shenzhen Key Laboratory of Energy Electrocatalytic Materials, Guangdong Research Center for Interfacial Engineering of Functional Materials, College of Materials Science and Engineering, Shenzhen University, Shenzhen, China. [2]College of Physics and Optoelectronic Engineering, Shenzhen University, Shenzhen, China. [3]College of Chemistry and Environmental Engineering, Shenzhen University, Shenzhen, China. [4]State Key Laboratory of Catalysis, Dalian Institute of Chemical Physics, Chinese Academy of Sciences, Dalian, China. [5]Global Institute of Future Technology, Shanghai Jiaotong University, Shanghai, China. ✉e-mail: lingyan.jing@foxmail.com; yangjl18@szu.edu.cn

challenge for improving catalytic activity, selectivity, and designing novel catalysts, as well as elucidating reaction mechanisms[20,21].

Colloidal carbon spheres are preferred for constructing parameter-tunable nanoreactors due to their unique advantages, such as adjustable porous structures, controllable particle size, large surface area, and adjustable surface chemistry[22–24]. Specifically, innovative synthetic methodology tends to provide carbonaceous structures with a wide range of tunability for active sites, such as oxygen-containing functional groups[25,26], defects[27,28], and edge sites[29,30]. The properties exhibited by carbon spheres present novel opportunities for screening appropriate probe reactions, such as the electrochemical two-electron oxygen reduction reaction (2e$^-$ ORR) for hydrogen peroxide ($H_2O_2$) production, and for exploring the operational mechanisms of nanoreactors[31,32].

Understanding the operational mechanisms of nanoreactors may drive the on-site electrosynthesis of practical $H_2O_2$ solutions as an oxidant, disinfectant, and emerging energy fuel (Fig. 1a)[33–35]. In addition, sustainable $H_2O_2$ electro-production eliminates the multi-step, high-energy input, and environmental impacts associated with the traditional anthraquinone (AQ) industry[36]. Efficient electrosynthesis of $H_2O_2$ relies on electrocatalysts that facilitate the highly active 2e$^-$ pathway[37]. Among the reported electrocatalysts, carbon-based materials have emerged as a promising alternative due to their abundance, low cost, and tunable catalytic properties[38]. To achieve high 2e$^-$ ORR performance, the surface of carbon material must effectively activate $O_2$ and maintain a suitable binding energy to *OOH intermediates[39,40]. Additionally, the carbon structure should possess a diffusion-friendly geometry to enable rapid separation of the generated $H_2O_2$ from the catalyst layer, preventing its further electro-reduction[41,42]. These challenges pose significant obstacles to the practical implementation of carbon materials for electrochemical 2e$^-$ ORR. Nevertheless, by employing nano-engineering techniques to manipulate the structure of carbon sphere materials and harnessing the catalytic functions, including diffusion and microenvironmental modulation within nanoreactors, there is a promising opportunity to overcome these challenges and advance practical strategies for $H_2O_2$ electrosynthesis.

In this work, we proposed the utilization of meticulously controlled carbon spheres as the fundamental building material and employed electrochemical 2e$^-$ ORR as a diffusion-related probe reaction to validate the catalytic functionality and superior performance of MHNs. The investigation began with finite element simulations (FES) to model the fluid behavior within the mesopores and hollow spaces of MHNs. Through precise tailoring of the internal structures of carbon spheres, a series of carbon-based MHNs was constructed as a proof of concept. The electrochemical properties and simulation results confirmed the enhanced diffusion and microenvironment-modulating effects of MHNs, particularly when designed with an appropriate hollow size. Leveraging these catalytic functionalities of the nanoreactor, the optimized MHCS$_{0.5}$ catalysts exhibited impressive activity (−3.1 mA cm$^{-2}$ at pH 13; −2.8 mA cm$^{-2}$ at pH 7) and selectivity (>95% at pH 13; >85% at pH 7) in $H_2O_2$ electrosynthesis conducted by rotating ring-disk electrode method. Moreover, the successful production of medical-grade $H_2O_2$ disinfectant in a flow cell device highlighted the significant practical potential of nanoreactor engineering. Therefore, our work can serve as a paradigm for advancing the design and understanding of nanoreactor construction and its application in the practical electrochemical production of $H_2O_2$ solutions.

## Results

### Optimizing diffusion-related geometrical parameters of MHNs
We endeavored to employ engineered carbon spheres to construct MHNs for the purpose of achieving efficient $H_2O_2$ electrosynthesis.

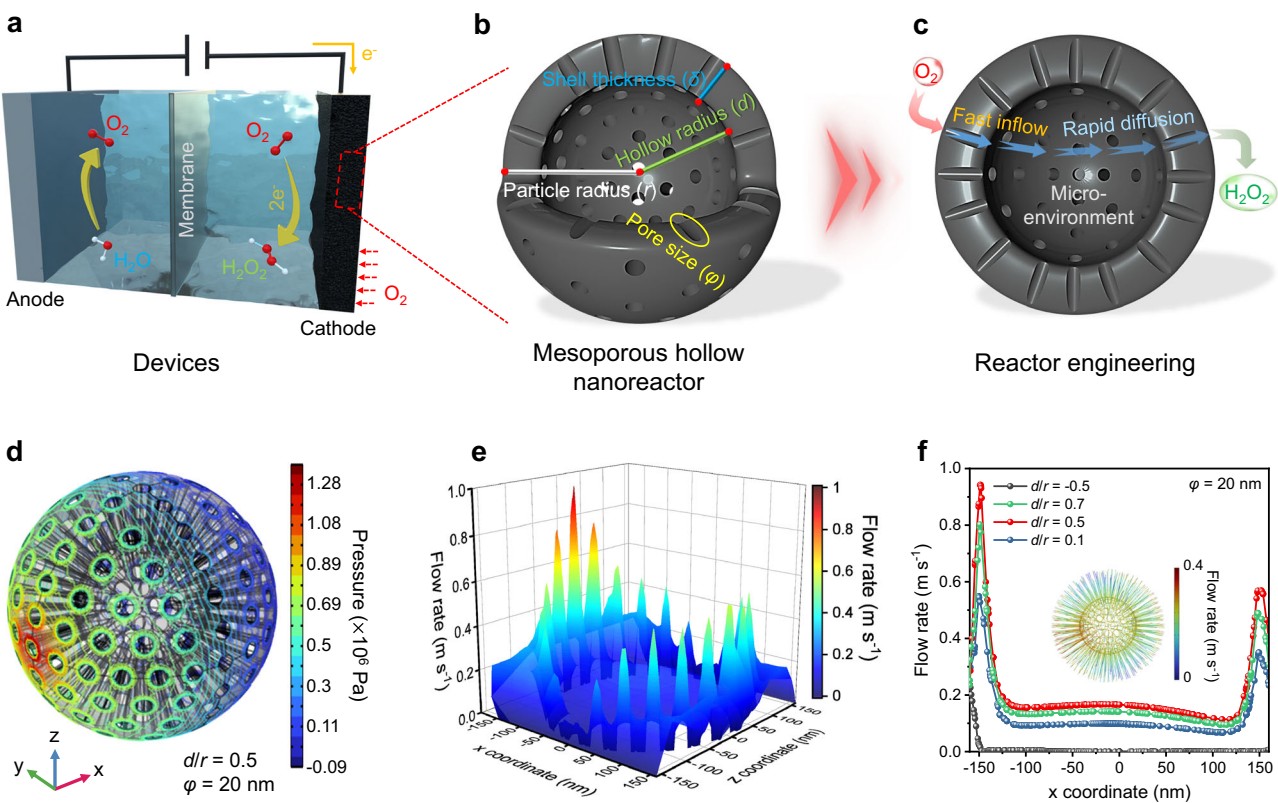

**Fig. 1 | Optimization of diffusion-related geometric parameters in MHNs.** **a–c** Schematic diagram of nanoreactor design based on carbon spheres for electrochemical 2e$^-$ ORR to produce $H_2O_2$. **d** Spatial distribution of pressure contours for the mesoporous carbon sphere model ($d/r = 0.5$, $r = 150$ nm, $\varphi = 20$ nm). **e** 3D color mapping of the spatial flow velocity distribution on the cross-section. **f** Fluid velocity distribution across mesoporous hollow carbon sphere models ($\varphi = 20$ nm) with varying hollow percentage ($d/r = -0.5$, 0.1, 0.5, and 0.7). Inset in **f** shows flow linear distribution within the hollow mesoporous sphere model.

Such MHNs typically consist of a mesoporous shell layer and an internal hollow, which can be calibrated with several crucial geometric parameters including particle radius ($r$), pore size ($\varphi$), shell thickness ($\delta$), and hollow radius ($d$) (Fig. 1b). These structural parameters may alter the fluid dynamics within MHNs, thereby regulating local diffusion and microenvironment, and consequently influencing the reaction process (Fig. 1c).

To investigate the correlation between the nanoreactor architecture and the fluid behavior inside, FES was conducted[43,44]. To ensure systematic construction of the MHNs and conserve computational resources for investigating the regularity of the electrolyte fluid, we standardized the particle size to 300 nm while maintaining the principle of unique variability for different $d/r$ or $\varphi$. As shown in Supplementary Fig. 1a, b, a mesoporous hollow sphere model ($d/r = 0.5$, $r = 150$ nm, $\varphi = 20$ nm) was constructed in three-dimensional (3D) square space (320 nm × 320 nm × 320 nm). To simulate fluid flow in an agitated system, the inlet and outlet of the fluid field were set as the left and right sides, respectively (Supplementary Fig. 1c, d). Such a setup generates a decreasing pressure gradient from left to right that drives water to flow spontaneously through the constructed geometry (Supplementary Fig. 2)[13]. Interestingly, there is a noticeable pressure drop inside the mesopores ($\varphi = 20$ nm) relative to the surface of the non-porous sphere (Supplementary Figs. 3 and 4). The pressure difference across the mesopores drives water flow through the channels and into the sphere interior, greatly expanding the available catalytic surface by increasing the contact between the electrolyte and carbon surface (Fig. 1d and Supplementary Fig. 5). More interestingly, driven by this pressure difference, a significant fluid acceleration appears in the mesoporous channels ($\varphi = 20$ nm), implying an intensified diffusion in MHNs (Supplementary Figs. 6 and 7). The 3D mapping of fluid velocity in the central cross-section ($y = 0$) confirms the local diffusion intensification within mesoporous channels and indicates the potential for rapid delivery of $O_2$ and exhaustion of generated $H_2O_2$ (Fig. 1e)[13].

To evaluate the diffusion effect and identify the optimal solution, a range of pore size parameters ($\varphi = 0$, 2 nm, 5 nm, 10 nm, 20 nm, and 30 nm) were simulated, along with calculating the fluid velocity distribution at specific locations (−160, 0, 0 → 160, 0, 0) to facilitate comparison and analysis (Supplementary Fig. 6b). The highest flow rates were observed in both the inflow and outflow mesopore channels at a pore size of 20 nm, despite the slower flow rate in the mesopore-linked hollow space (Supplementary Fig. 8–10). Based on the above, we continue to maintain similar particle parameters but vary the hollow ratios for predicting fluid mass transfer. So, models with different hollow occupancy ratios ($d/r = -0.5$, 0.1, 0.5, and 0.7) are also simulated (Supplementary Figs. 9 and 11). The results indicate that a certain size of hollow space is crucial, with the maximum flow velocity in the mesoporous channels occurring at $d/r = 0.5$ (Fig. 1f). Considering that the active sites for the electrochemical ORR are dispersed on the carbonaceous surface of the mesoporous shells, the localized diffusion enhancement enables the swift inflow of $O_2$ and the rapid transfer of the generated $H_2O_2$ into the bulk solution, thereby averting its accumulation on the catalyst surface and subsequent electro-reduction[45]. These findings indicate that the density of mesopores may influence the fluid behavior within the MHNs. For a single MHN, a small $d/r$ ratio suggests a long mesoporous channel and high mesoporous density, resulting in continuous viscous resistance that affects the flow rate[13,46]. Conversely, a large $d/r$ implies a short mesoporous channel and a larger hollow structure with lower mesoporous density, enabling rapid fluid entry into the hollow's flow rate buffer zone, though limiting fluid acceleration. Additionally, the modified MHN model, constructed with reduced mesopore density, demonstrates the electrolyte's ability to maintain a high flow rate over a long distance (Supplementary Fig. 12). However, the decrease in mesoporous channels would cause most of the electrolyte to flow from the MHN's surface rather than through the mesopores, potentially reducing the utilization of the internal active

sites. The fluid buffering effects in the hollow region may establish a potentially stable internal microenvironment. To summarize, micro-mechanical dynamics simulations were employed to identify diffusion-favorable structural parameters ($d/r = 0.5$, $\varphi = 20$ nm) and guide the design of MHNs.

## Refinement and characterization of carbon-based MHNs

Inspired by the simulation results above, a sequential organic-inorganic hybridization co-assembly approach was developed to fabricate mesoporous hollow carbon sphere-based nanoreactors. As shown in Supplementary Fig. 13, this approach begins with the chronological modular assembly of organic phenolic resin and inorganic silica ($SiO_2$)[47,48]. In detail, resorcinol and formaldehyde are polymerized under the catalyst of ammonia to form phenolic resins, while tetrapropoxysilane (TPOS) added simultaneously underwent hydrolysis to form $SiO_2$. And then, these two components spontaneously co-assemble into the hybrid resin/$SiO_2$ spheres (Supplementary Fig. 14a, b). After further pyrolysis of resin/silica spheres in an inert atmosphere, the resin zone is converted into carbon components, resulting in the formation of C/$SiO_2$ spheres (Supplementary Fig. 14c, d). The carbonaceous material in the C/$SiO_2$ spheres was selectively eliminated by calcination under an air atmosphere, yielding mesoporous silica spheres (MSiO₂S) with a solid core, providing compelling evidence that TPOS was initially hydrolyzed to form a $SiO_2$ core, which was subsequently surrounded by the hybridization of resin and $SiO_2$ (Supplementary Fig. 15). Thus, the selective removal of $SiO_2$ component from the C/$SiO_2$ spherical precursor led to the formation of mesoporous hollow carbon spheres (MHCS$_x$, x = $d/r$) by generating mesopores and hollow in the areas previously occupied by the $SiO_2$ phase[49]. The morphology of the MHCS$_x$ was analyzed by SEM and TEM, revealing an average particle radius of 210 nm and a hollow radius of 105 nm, resulting in the designation of MHCS$_{0.5}$ (Fig. 2a–e). Further, high-resolution transmission electron microscopy (HRTEM) indicated the permeable mesoporous channels on the external shell (Fig. 2f). Besides, some mesoporous channels running through the shell layer were clearly observed in both the SEM images of the crushed MHCS$_{0.5}$ and the locally enlarged TEM images of MHCS$_{0.5}$, providing a path for the electrolyte fluid to pass through the internal hollow space (Supplementary Fig. 16). Additionally, the high-angle annular dark-field scanning transmission electron microscopy (HAADF-STEM) image confirms the hollow mesoporous architecture of MHCS$_{0.5}$, as well as the even distribution of carbon and oxygen elements (Fig. 2g, h). To confirm the flow of electrolyte through the internal hollow space of MHCS$_{0.5}$, we conducted an initial one-hour ORR using MHCS$_{0.5}$ in 0.5 M KCl electrolyte. Following the reaction, the MHCS$_{0.5}$ samples were dried without washing for direct TEM observation, revealing the presence of residual KCl components filling the interiors of MHCS$_{0.5}$ (Supplementary Fig. 17). These findings are further validated by the corresponding elemental linear scans, confirming the presence of mesoporous channels running through the shell layers of MHCS$_{0.5}$ and the flow of electrolyte fluids through these channels, enabling movement of electrolyte through the hollow interior (Supplementary Fig. 18).

In the sequential organic-inorganic hybridization co-assembly strategy, the timing of resin and $SiO_2$ assembly can be flexibly manipulated to significantly adjust the size of the hollow structure of MHCS. We attempted to manipulate the time nodes on formation of resin and $SiO_2$ to synthesize mesoporous spheres with varying hollow sizes. To achieve this, we added resin monomers (resorcinol and formaldehyde) first, followed by the addition of TPOS after a specific time interval ($t$ min). For the synthesis of MHCS$_{0.5}$, the value of $t$ was controlled to be 0, implying that the resin monomers and the TPOS required for hydrolysis to form $SiO_2$ were added simultaneously. If TPOS is added 20 min earlier ($t = -20$), the hydrolysis process of TPOS

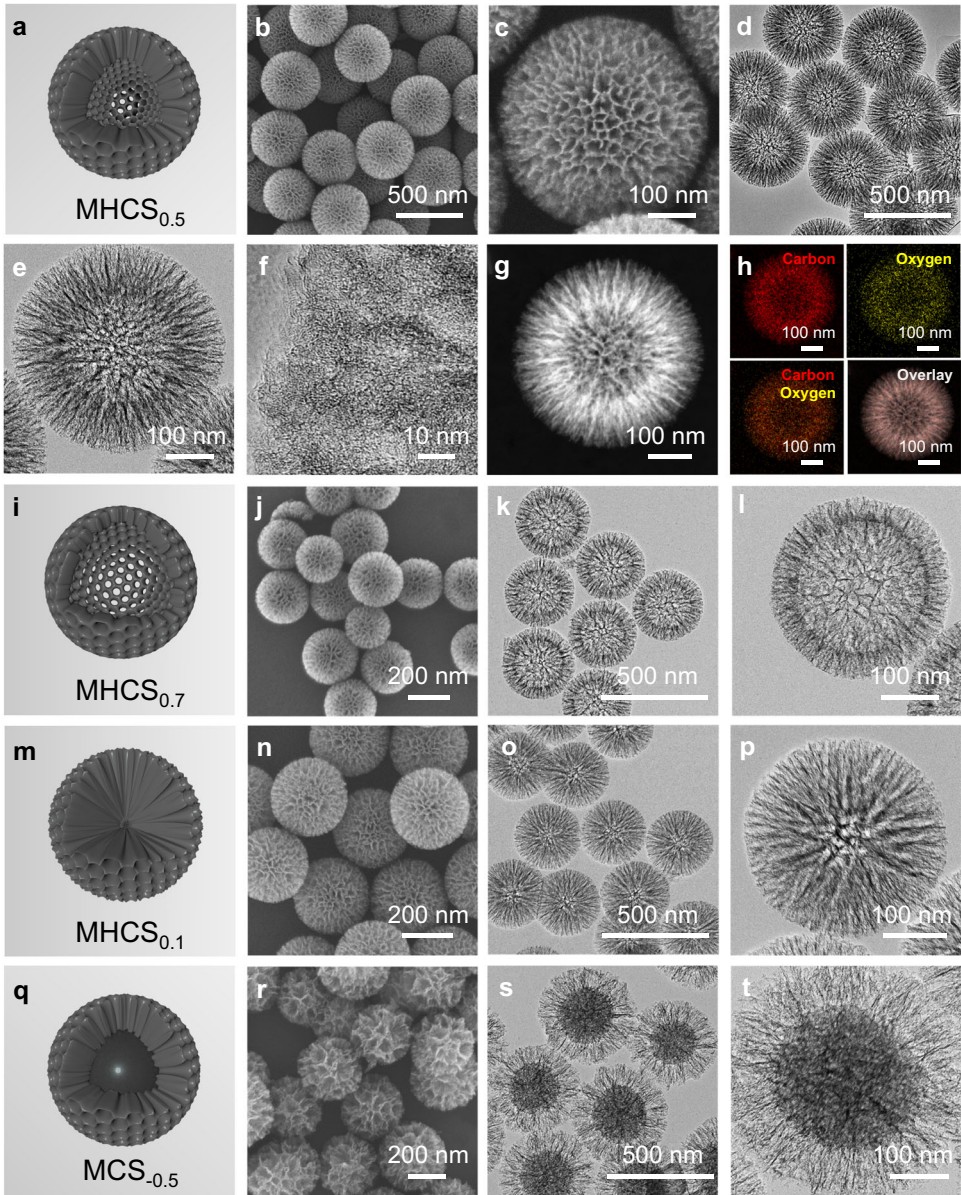

**Fig. 2 | Electron microscopy images of synthesized carbon-based MHNs.**
**a** Structural model, **b, c** SEM images, **d, e** TEM images, **f** HRTEM image, **g** HAADF-STEM image, and the corresponding **h** elemental mapping images of MHCS$_{0.5}$.

**i** Structural model, **j**, SEM image, and **k, l** TEM images of MHCS$_{0.7}$. **m** Structural model, **n** SEM image, and **o, p** TEM images of MHCS$_{0.1}$. **q** Structural model, **r** SEM image, and **s, t** TEM images of MCS$_{-0.5}$.

is advanced, leading to the formation of larger SiO$_2$ solid core before it is hybridized with the resin. After carbonization and SiO$_2$ removal, MHCS$_{0.7}$ with larger hollow size than MHCS$_{0.5}$ was synthesized (Fig. 2i–l). By delaying the addition of silane by 20 min ($t = 20$), the hybridization co-assembly of SiO$_2$ with resin can be advanced correspondingly, resulting in the formation MHCS$_{0.1}$ with an ultra-small hollow (Fig. 2m–p). Further, by adding the resin monomer for 80 min and then hydrolyzing TPOS ($t = 80$), a resin solid core is first polymerized and then surrounded by a hybridization of SiO$_2$ and resin, resulting in the formation of mesoporous carbon spheres (named MCS$_{-0.5}$) with solid carbon cores and mesoporous carbon shells (Fig. 2q–t). The geometrical parameters of the synthesized series of MHCS demonstrated a clear positive correlation between the ratio of hollow size to particle size ($d/r$) and the $t$ value, providing evidence for the controlled synthesis of hollow mesoporous nanoreactors under time order manipulations (Supplementary Table. 1 and Supplementary Fig. 19).

Some comparative experiments were also conducted to confirm the role of SiO$_2$, derived from the hydrolysis of TPOS, in the formation of hollow mesoporous carbon spheres. Without the addition of TPOS, only the polymerization of phenolic resin would occur, resulting in the formation of non-mesoporous resin spheres (Supplementary Fig. 20). Pyrolysis of these resin spheres would then produce microporous carbon spheres (CS, Supplementary Fig. 21). Additionally, by burning off the resin components of resin/SiO$_2$ spheres assembled at various time intervals ($t = -20$, 20 and 80) in the air atmosphere, various mesoporous spherical SiO$_2$ structures with the topologies opposite to that of the corresponding MHCS$_x$ (MHCS$_{0.7}$, MHCS$_{0.1}$ and MCS$_{-0.5}$) were obtained (Supplementary Fig. 22–24). These findings substantiate the controlled manipulation of hierarchical heterogeneous assembly of SiO$_2$ and resin in a specific temporal order.

We analyzed the porous structure and physicochemical characteristics of five samples: MCS$_{-0.5}$, MHCS$_{0.1}$, MHCS$_{0.5}$, MHCS$_{0.7}$, and

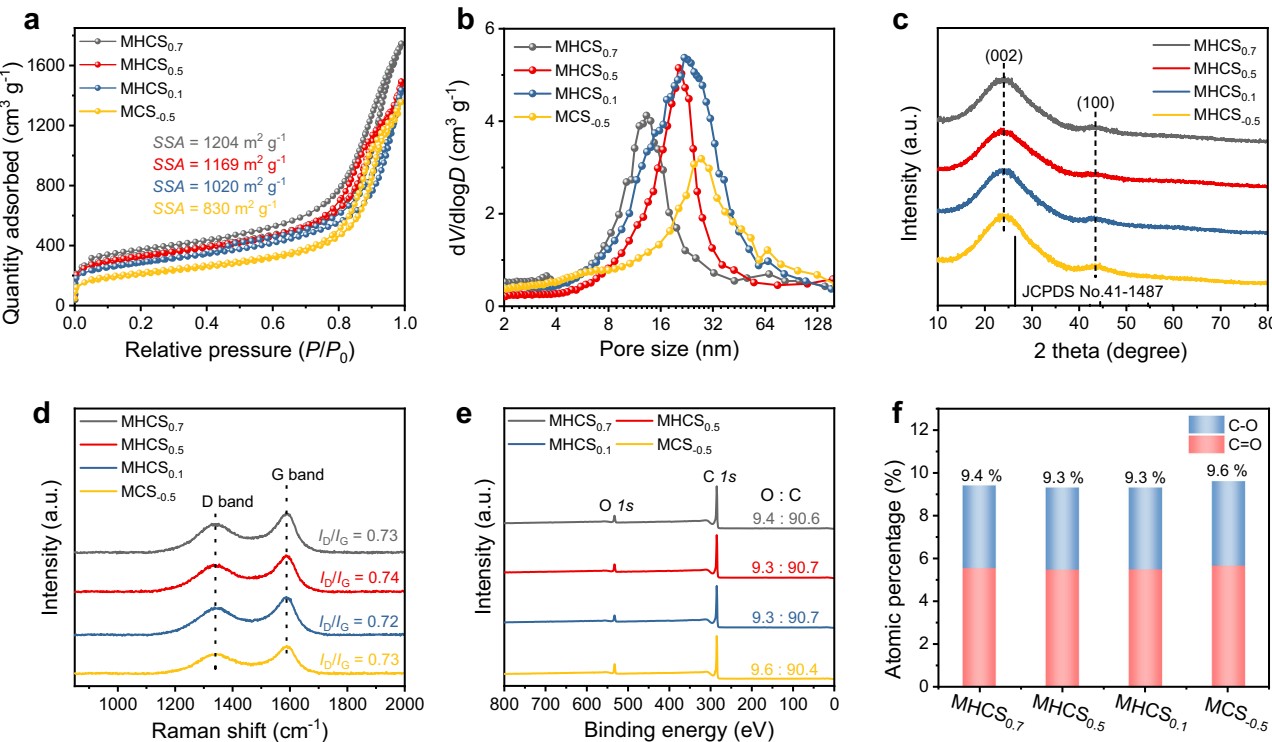

**Fig. 3 | Characterization of as-obtained MHNs. a** Nitrogen adsorption-desorption isotherms, **b** pore size distribution curves, **c** XRD patterns, **d** Raman spectra, **e** XPS survey spectra, and the corresponding **f** C-O/C = O species content of MHCS$_x$ samples.

CS. The nitrogen adsorption and desorption isotherms revealed high Brunauer-Emmett-Teller (BET) specific surface areas (SSA) of 830, 1020, 1169, and 1204 m$^2$ g$^{-1}$ for MCS$_{-0.5}$, MHCS$_{0.1}$, MHCS$_{0.5}$, and MHCS$_{0.7}$, respectively (Fig. 3a). The Barrett-Joyner-Halenda method revealed the mesopore size distribution of MHCS$_x$ samples to be in the range of 15–25 nm (Fig. 3b). Compared to MHCS$_x$, CS has a lower specific surface area of only 552 m$^2$ g$^{-1}$ and almost no mesopore distribution (Supplementary Fig. 25). XRD patterns of MHCS$_x$ and CS showed broad peaks at 23.0° and 43.6°, corresponding to the (002) and (100) planes of the disordered carbonaceous structure, respectively (Fig. 3c and Supplementary Fig. 26a)[50]. The Raman spectra showed similar intensity ratios of D and G bands and a G linewidth of 100 cm$^{-1}$ for all five samples, indicating typical characteristics of amorphous carbon with comparable defects (Fig. 3d and Supplementary Fig. 26b)[51]. XPS analysis revealed an oxygen content of around 9.5 at% for MHCS$_x$ (Fig. 3e). The C 1s spectrum shows three peaks at 284.6, 286.0, and 288.7 eV, corresponding to C − C, C − O, and O − C = O, respectively, while the O 1s spectrum exhibits two peaks assigned to C = O (531.6 eV) and C − O (533.0 eV) (Supplementary Fig. 27 and Supplementary Fig. 28)[43]. Fig. 3f and Supplementary Table. 2 shows that the distribution of oxygen-containing functional groups, which are considered as active sites for electrochemical 2e$^-$ ORR[25,26,52,53], is similar among the different MHCS$_x$ samples. Besides, CS had a similar chemical composition to MHCS$_x$ (Supplementary Fig. 29). In addition, defective carbon or some carbon edges, have been proposed as potential active sites for 2e$^-$ ORR in some previous studies[30,54]. To examine the defective carbon, we utilized electron paramagnetic resonance (EPR) and observed nearly identical signal intensities on MHCS$_x$, suggesting that MHCS$_x$ exhibits a consistent level of defectivity for electrochemical 2e$^-$ ORR (Supplementary Fig. 30). These similar properties observed in MHCS$_x$ may be attributed to their shared origin as resorcinol-formaldehyde resin-derived carbon and undergoing the same carbonization process. Therefore, these as-obtain carbon spheres have similar chemical properties and active sites, with differences mainly in hollow size and thickness of

mesoporous shells, making them ideal models for exploring the efficacy of MHNs for catalytic reactions.

**Electrocatalytic performance**

As a proof of concept, the electrochemical 2e$^-$ ORR was used as a probe reaction to evaluate catalytic process intensification due to fluid transport in this series of carbon-sphere based MHNs. The electrochemical performance was evaluated using a three-electrode rotating ring-disk electrode (RRDE) technique at 1600 rpm in an O$_2$-saturated electrolyte, with the Pt ring electrode set at 1.2 V vs reversible hydrogen electrode (RHE) to detect the produced H$_2$O$_2$ on the disk. Remarkably, the MHCS$_{0.5}$ achieves a large diffusion-limiting disk current density (−3.1 mA cm$^{-2}$ at 0.20 V vs RHE) in 0.1 M KOH (pH = 13), approaching the theoretical limit for 2e$^-$ ORR (Fig. 4a)[41,55]. Besides, the MHCS$_{0.5}$ delivers a high onset potential of 0.85 V vs RHE, larger than the thermodynamic equilibrium 2e$^-$ ORR potential (0.75 V)[52,56,57]. The high ORR onset potential may stem from its appropriate porous structure and geometric framework, which enhance contact with electrolyte fluids to improve the utilization of carbon surface active sites in the electrolyte environment[14,19,58]. Correspondingly, the electrochemical activity of non-mesoporous CS was found to be poor, as indicated by a very low disk current densities at diffusion-limiting conditions, suggesting the insufficient the exposure of active sites in the micropores (Supplementary Fig. 31). Figure 4b, c shows the calculated H$_2$O$_2$ selectivity and the corresponding electron transfer number ($n$) for MHCS$_x$ samples. The MHCS$_{0.5}$ catalyst delivers the high H$_2$O$_2$ selectivity (> 95%) and a $n$ value around 2.05 across a wide potential window (-0.5 V, from 0.3 V to 0.8 V vs RHE), which are superior to those of most of the previously reported catalysts (Supplementary Table. 3). Decreases in H$_2$O$_2$ selectivity were observed in the order of MHCS$_{0.5}$, MHCS$_{0.7}$, MHCS$_{0.1}$, and MCS$_{-0.5}$, showing a positive relationship with the simulated flow rate in the mesopore channels of the corresponding MHNs models.

To comprehend the distinctions in the electrochemical properties of MHCS$_x$, it is imperative to uncover the intrinsic active sites involved

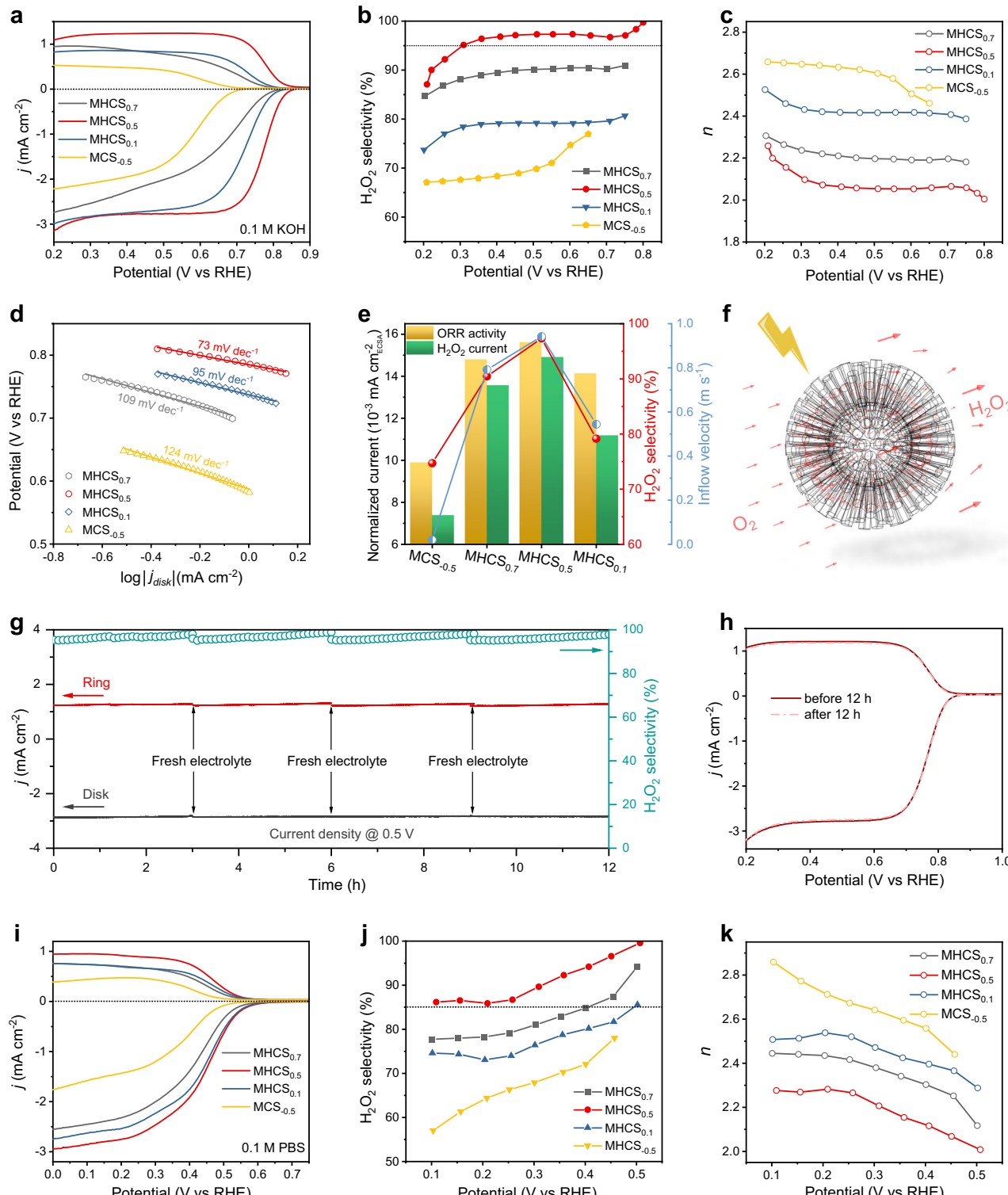

**Fig. 4 | Electrochemical performance of MHNs. a** LSV curves of RRDE measurements at 1600 rpm in 0.1 M KOH, **b** the corresponding $H_2O_2$ selectivity, and **c** electron transfer number at various applied potentials. **d** Tafel plots derived from LSV curves in 0.1 M KOH. **e** Normalized electrochemical activity and $H_2O_2$ partial current on $MHCS_x$ (@ 0.6 V vs RHE), along with the corresponding $H_2O_2$ selectivity and electrolyte inflow velocity. **f** Schematic diagram depicting the flow of fluid/ oxygen and the oxygen reduction process on the $MHCS_{0.5}$ model. **g** Stability measurements of $MHCS_{0.5}$ at a fixed disk potential of 0.5 V vs RHE in 0.1 M KOH. **h** LSV curves of $MHCS_{0.5}$ before and after 12 h stability test. **i** LSV curves of RRDE measurements at 1600 rpm in 0.1 M PBS, **j** the corresponding $H_2O_2$ selectivity, and **k** electron transfer number at various applied potentials.

in electrocatalytic ORR. We reduced the $MHCS_{0.5}$ sample in a mixed $H_2/N_2$ atmosphere to eliminate the oxygen component, resulting in R-$MHCS_{0.5}$ with an oxygen content as low as 1.76 at% (Supplementary Fig. 32a). Consequently, the ORR activity and $H_2O_2$ selectivity of R-$MHCS_{0.5}$ were significantly reduced (Supplementary Fig. 32b), indicating that the oxygen species on the carbon matrix serve as the active source of 2e⁻ ORR, consistent with some previous reports[52,59]. Given the comparable oxygen species on $MHCS_x$, the notable discrepancy in

$H_2O_2$ selectivity is attributed to the fluid behavior of electrolytes within different architectures of MHNs. Fig. 4d displays the Tafel slopes of $MHCS_x$, with $MHCS_{0.5}$ exhibiting a low value of 73 mV $dec^{-1}$, indicating its fast reaction kinetics. Tafel plots reflect various rate-determining steps (RDS), and a progressive decrease in Tafel slopes could indicate a shift in the RDS from the first electron-transfer step ($O_2 \rightarrow$ *OOH, -120 mV $dec^{-1}$) to the last $H_2O_2$ desorption step (*OOH $\rightarrow H_2O_2$, ~ 70 mV $dec^{-1}$)[60,61]. The ORR pathway was studied from the perspective of the $H_2O_2$ reduction reaction ($H_2O_2RR$) activity in $N_2$-saturated 0.1 M KOH containing 10 mM $H_2O_2$. Among $MHCS_x$, $MHCS_{0.5}$ exhibits apparent inertness to $H_2O_2RR$, with negligible currents, confirming its highly selective 2e⁻ pathway (Supplementary Fig. 33). In contrast, we observed an increasing trend in $H_2O_2RR$ current from $MHCS_{0.7}$, $MHCS_{0.1}$ to $MCS_{-0.5}$, suggesting an improved $H_2O_2RR$ capability. Considering the simulated fluid velocity on $MHCS_x$, we deduce that the low flow rates would advance the indirect 4e⁻ pathway to reduce the $H_2O_2$ selectivity[61,62].

These carbon sphere-based electrocatalyst were further evaluated for their accessible active surface area (ECSA) using a double-layer capacitance ($C_{dl}$) method. Specifically, the $C_{dl}$ values for $MHCS_{0.5}$, $MHCS_{0.1}$, $MHCS_{0.7}$, and $MCS_{-0.5}$ were calculated to be 8.04, 7.20, 4.36, and 3.17, respectively (Supplementary Fig. 34, 35). Interestingly, the $C_{dl}$ values of $MHCS_x$ do not align with their total BET surface area, but instead, exhibit a positive correlation with the external surface area excluding the microporous region (Supplementary Fig. 35b)[63]. Considering the particle morphology of $MHCS_x$, it can be hypothesized that the internal surface exposure, facilitated by the mesoporous channels, enhances the interaction between the electrolyte and active sites, thereby promoting heightened electrocatalytic activity[64,65]. Prominently, the high $C_{dl}$ value on $MHCS_{0.5}$ signifies its structurally favorable diffusion, suggesting a swift supply of electrolyte ions and effective utilization of active sites. Besides, the non-mesoporous CS exhibits a low $C_{dl}$ of 2.38 mF $cm^{-2}$, further illustrating the restricted accessibility of micropores by electrolytes (Supplementary Fig. 36)[54]. Furthermore, we utilized ECSA to standardize the electrochemical activity and the $H_2O_2$ partial currents of $MHCS_x$ at diffusion-related potentials (0.6 V vs RHE), as illustrated in Fig. 4e. Interestingly, the associated normalized currents and 2e⁻ selectivity exhibit a positive correlation with the simulated electrolyte inflow velocity, providing additional evidence for the connection between the diffusion effect and the electrochemical performance on these MHNs. These characteristics demonstrate the influence of the hollow's structural parameters on diffusion and their reflection on electrocatalytic performance (Fig. 4f).

To assess the catalytic stability, we conducted chronoamperometric tests for $MHCS_{0.5}$ using the RRDE technique at a constant disk potential of 0.5 V vs RHE. During the continuous testing, the accumulated $H_2O_2$ might lead to a slight rise in the ring current and subsequently in selectivity. To maintain a consistent ring current, we replaced the electrolyte every 3 h and performed electrochemical cleaning of the Pt ring to remove any effects of the accumulated $H_2O_2$ during continuous operation. The results showed that the $MHCS_{0.5}$ catalyst maintained stable disk/ring currents and achieved over 95% selectivity throughout the 12-hour continuous test (Fig. 4g). Further, we conducted an accelerated durability test (ADT) on $MHCS_{0.5}$ by sweeping the potential between 0.2 and 1.0 V vs RHE for 12 h. The similar linear sweep voltammetry curves obtained before and after ADT confirm its high stability (Fig. 4h). After long-term electrolysis, $MHCS_{0.5}$ still maintains its hollow mesoporous structure, and its chemical composition remains consistent with its pre-reaction state, ensuring its stability and reusability (Supplementary Figs. 37 and 38). Of particular interest, $MHCS_{0.5}$ demonstrates exceptional 2e⁻ ORR performance in neutral electrolyte conditions (pH 7), with a diffusion-limited disk current of −2.8 mA $cm^{-2}$ at 0.2 V vs RHE, an onset potential of 0.6 V vs RHE, and an $H_2O_2$ selectivity exceeding 85%, surpassing vast

majority of carbon-based catalysts reported in the literature (Fig. 4i–k and Supplementary Table. 4). The trends observed in the ORR activity and selectivity of $MHCS_x$ under neutral conditions were consistent with those observed under alkaline conditions. Interestingly, the polarization curves exhibit two distinct sets of slopes at different potentials, suggesting variations in the ORR mechanism. Consistently, the $H_2O_2RR$ test conducted on $MHCS_{0.5}$ revealed a low $H_2O_2RR$ current density, indicating its overall inertness to $H_2O_2RR$ in a neutral electrolyte, with a slight increase observed at more negative potentials (Supplementary Fig. 39). This trend in the $H_2O_2RR$ current aligns with that of the ORR, highlighting the influence of the indirect 4e⁻ process at relatively high voltages on the 2e⁻ selectivity[61].

## Actual $H_2O_2$ electrosynthesis properties

The practical application of $MHCS_{0.5}$ catalysts was evaluated using a H-type cell setup in 0.1 M KOH with continuous $O_2$ bubbling (Supplementary Fig. 40a). The produced $H_2O_2$ was quantified by the $Ce^{4+}$ titration method with the ultraviolet-visible (UV–vis) spectrophotometric technique. During a long-term stability test at a fixed voltage of 0.5 V vs RHE, the current remains stable at about 5.4 mA $cm^{-2}$ and the concentration of accumulated $H_2O_2$ increases continuously, resulting in 40.29 mmol $L^{-1}$ within 12 h (Supplementary Fig. 40b). During the long-term continuous test, the fluctuations of current and Faraday efficiency (FE) (> 97%) are slim, indicating the reliable practicality of the bulk $MHCS_{0.5}$ electrocatalyst.

To address the limitations of peroxide generation in a static H-cell configuration, characterized by inefficient mass transport due to low $O_2$ diffusion efficiency and $H_2O_2$ desorption rate, leading to suboptimal apparent productivity,[66,67] we further evaluated the actual $H_2O_2$ production from $MHCS_{0.5}$ using a flow cell setup (Fig. 5a and Supplementary Fig. 41). In the measurements, a $MHCS_{0.5}$ loaded carbon paper (CP) electrode, including a gas diffusion layer (GDL), was used as the cathode, and a commercial $IrO_2$ served as the anode in the flow cell. As depicted in Fig. 5b, $MHCS_{0.5}$ exhibited a sharp increase in current density with rising voltage, reaching ~254 mA in 0.1 M KOH and ~188 mA in 0.1 M phosphate buffered solution (PBS) at an uncompensated voltage of 0.1 V vs RHE. Moreover, the current of $MHCS_{0.5}$ showed a consistent and steady profile during continuous electrolysis at an applied voltage of 0.1 V vs RHE (Supplementary Fig. 42). Notably, the production yield of $H_2O_2$ exhibited a steady increase over time, with a 16-hour cumulative yield of $H_2O_2$ reaching 68.7 mmol in 0.1 M KOH and 50.6 mmol in 0.1 M PBS, corresponding to $H_2O_2$ production rates of 17.18 and 12.64 mol $g_{catalyst}^{-1}$ $h^{-1}$ (Fig. 5c, d) and outperforming most recently reported electrocatalysts (Supplementary Table. 5). Additionally, the Faraday efficiency of $H_2O_2$ electrosynthesis on $MHCS_{0.5}$ remained consistently around 90% throughout a long-term test. After 16 h of electrosynthesis in the flow cell, the cumulative concentration of $H_2O_2$ reached 5.84 wt.% in an alkaline environment and 4.29 wt.% in a neutral medium, satisfying the concentration requirements for the medical-grade disinfectant (~ 3 wt.%)[68]. Notably, the electroproduction of $H_2O_2$ solutions in neutral electrolytes is particularly appealing due to its environmental friendliness, minimal corrosiveness, and potential for reduced electrolytic cell costs[69]. Additionally, neutral $H_2O_2$ solutions provide a sustainable and versatile option for practical applications, including the direct use of synthesized $H_2O_2$ in biochemical systems[70,71]. The robust performance and excellent selectivity for 2e⁻ reduction demonstrated by $MHCS_{0.5}$ in the flow cell setup validate its potential for practical applications in electrochemical $H_2O_2$ production, particularly for the electro-production of neutral $H_2O_2$ solutions[37]. In contrast, $MCS_{-0.5}$ exhibited restricted electrochemical activity, with currents of 51 mA at 0.1 M KOH and 25 mA at 0.1 M PBS at an applied potential of 0.1 V vs RHE (Supplementary Fig. 43a). Additionally, the FE of $MCS_{-0.5}$ for $H_2O_2$ electroproduction is approximately 60%, significantly lower than that of the $MHCS_{0.5}$ electrode (Supplementary Fig. 43b, c). This noticeable

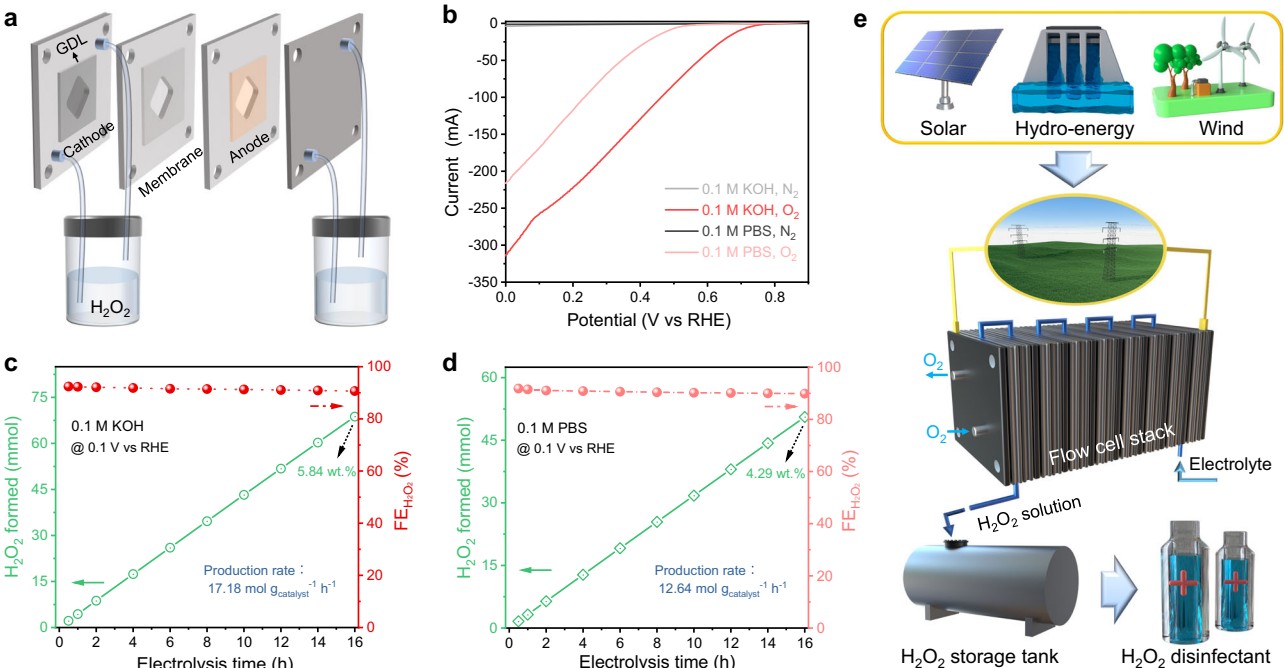

**Fig. 5 | Actual H₂O₂ electro-production based on MHNs. a** Schematic diagram of the flow cell for H₂O₂ electrosynthesis. **b** Polarization curves of MHCS₀.₅ loaded CP electrode in flow cell. **c** Electrolysis time-dependent H₂O₂ production and faradaic efficiency of MHCS₀.₅ under continuous O₂ purging in 0.1 M KOH. **d** Electrolysis time-dependent H₂O₂ production and faradaic efficiency of MHCS₀.₅ under continuous O₂ purging in 0.1 M PBS. **e** Envisioned green electricity-driven large-scale electrochemical production of H₂O₂ disinfectant.

distinction in electrochemical performance further underscores the effectiveness of the engineered carbon sphere-based MHNs.

Based on the remarkable activity, high $H_2O_2$ selectivity, and reliable stability of MHCS₀.₅, coupled with the global goal of carbon neutrality, we envision a future where large-scale production of $H_2O_2$ disinfectants is achieved using sustainable energy sources[67,72]. As depicted in Fig. 5e, surplus green electricity generated from solar, hydro, and wind energy sources is utilized as the energy supply for decentralized $H_2O_2$ production in an electrolytic cell stack. Significantly, the medical-grade $H_2O_2$ solution produced under neutral conditions can be elegantly stored or directly employed for disinfection and sterilization purposes. This green electricity-driven $H_2O_2$ disinfectant production represents a sustainable approach that enables on-site production and utilization of $H_2O_2$, circumventing the pollution, high energy consumption, and explosion hazards associated with the traditional AQ industry. In the pursuit of this goal, the nanoreactor engineered carbon spheres will serve as pivotal catalysts in realizing this vision.

## Microenvironmental modulation effect of nanoreactors

To gain a deeper understanding of the exceptional 2e⁻ ORR performance of MHCS₀.₅, particularly in neutral conditions, we conducted extensive simulations to elucidate the catalytic mechanism of the hollow mesoporous nanoreactor. To provide a visual representation of the interior within the nanoreactor, a two-dimensional (2D) model of a hollow mesoporous sphere ($d/r = 0.5$, $r = 150$ nm, $\varphi = 20$ nm) was constructed[49,73]. As shown in Supplementary Fig. 44, the mesoporous hollow sphere is modeled as having all surfaces as catalytically active sites and placed within a square simulation domain (320 nm × 320 nm), with the inflow and outflow boundaries for the electrolyte set at the left and right sides. Consistent with the results of the previous 3D simulations, the O₂-containing electrolyte fluid is driven by the pressure gradient to flow through the mesopore-hollow-mesopore of the sphere and ultimately exits the simulation domain (Supplementary Fig. 45). We obtained the velocity distribution at y = 0 and observed a

significant acceleration of the fluid flow in the mesopore region, while the flow velocity within the hollow region was relatively low (Supplementary Fig. 46). High flow rates can increase the entrainment of O₂ per unit time, as evidenced by the spatial concentration distribution of O₂ being almost proportional to the flow rate distribution, leading to the enrichment of O₂ in pore channels (Fig. 6a, b). From a reaction kinetics perspective, the local enrichment of O₂ in the mesoporous channels can promote the activation of O₂ (O₂ + * → *O₂) and stimulate the formation of *OOH as well as the consumption of protons to generate OH⁻ (*O₂⁻ + H₂O → *OOH + OH⁻)[67,74,75]. The accumulation of OH⁻ generated, which serves as a flow buffer, causes a local pH elevation within the MHNs, as confirmed by the finite element simulation results (Fig. 6c, d). After the electrolysis in neutral media (0.1 M PBS, pH = 7), phenolphthalein indicator was quickly dropped onto the MHCS₀.₅ electrode and inserted into a bottle containing neutral electrolyte. The appearance of a slight purplish red color on the electrolytic surface was observed, providing evidence for the presence of a localized pseudo-alkaline microenvironment (Supplementary Fig. 47). More directly, a significant in-situ color change was observed on MHCS₀.₅ electrode surface during a 2e⁻ ORR in neutral electrolyte (0.1 M K₂SO₄) containing phenolphthalein (Supplementary Fig. 48). This color change and the electrocatalytic process proceeded almost simultaneously, providing evidence of a local pH increase near the catalyst layer (Supplementary Movie 1). Next, we conducted the following experiments to verify the MHNs' ability to accumulate OH⁻. Initially, as shown in Fig. 6e, the ORR process was performed for 10 min in a neutral electrolyte using a carbon sphere catalyst at 0.4 V vs. RHE, with the corresponding current density recorded in Supplementary Fig. 49a. Subsequently, all electrodes were extracted from the cell, and the working electrode was rinsed by immersion in 5 mL of an aqueous solution. This process was repeated 10 times, and the pH of the aqueous solution was measured. Fig. 6f illustrates that the MHCS₀.₅ electrode surfaces exhibited higher OH⁻ concentrations compared to the other samples. This observation is associated with the high current density generated by the MHCS₀.₅ electrodes, leading to increased

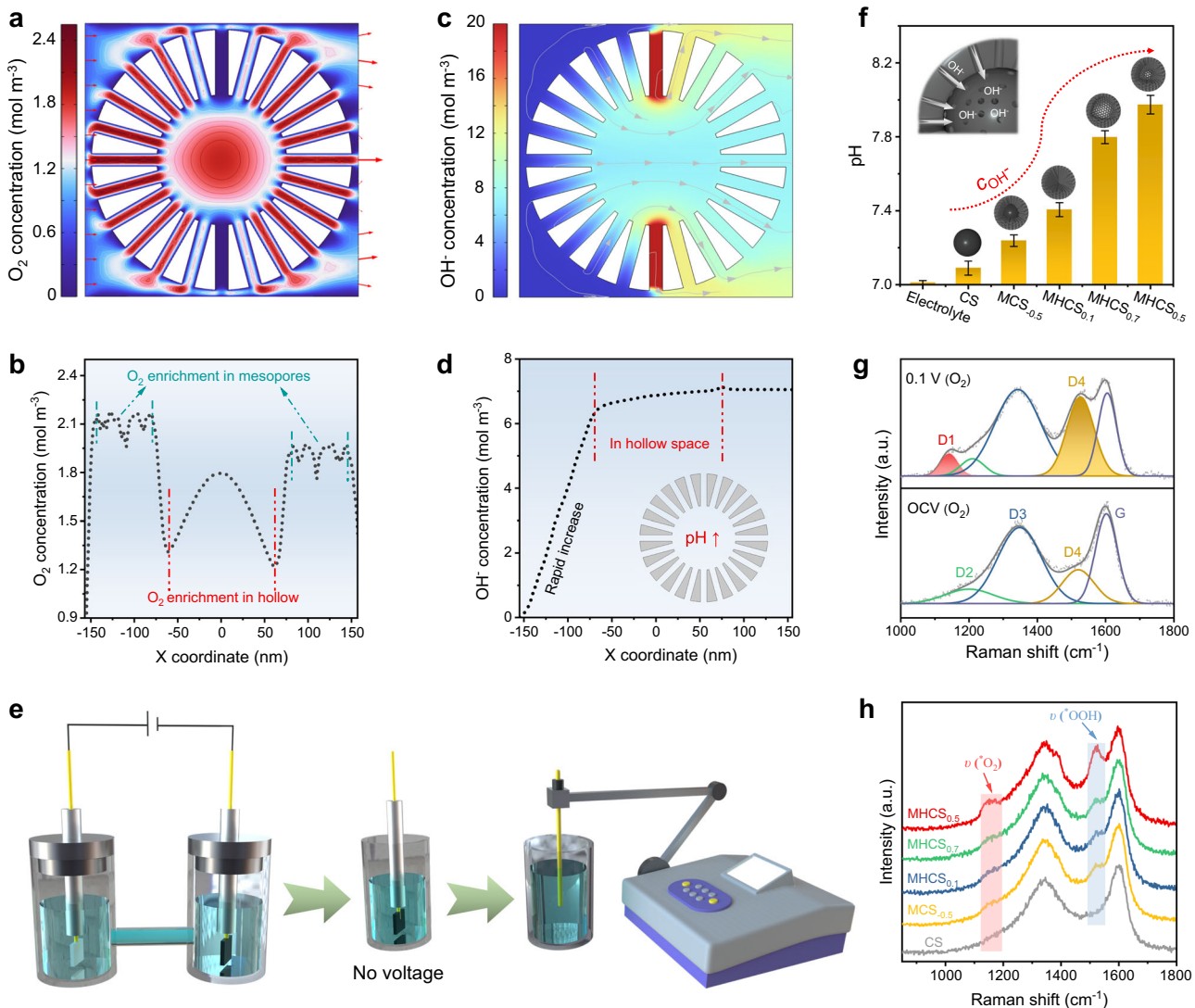

**Fig. 6 | Microenvironmental modulation in MHNs. a** Spatial distribution of $O_2$ concentration in the mesoporous carbon sphere model ($d/r = 0.5$, $r = 300$ nm, $\varphi = 20$ nm), and **b** the corresponding $O_2$ concentration distribution curve of mesoporous carbon sphere model on $y = 0$. **c** Spatial distribution of $OH^-$ concentration of the mesoporous carbon sphere model ($d/r = 0.5$, $r = 300$ nm, $\varphi = 20$ nm), and **d** the corresponding $OH^-$ concentration distribution curve of mesoporous carbon sphere model on $y = 0$. **e** Schematic diagram of the process for testing the $OH^-$ concentration on the electrode surface. **f** Comparison of $OH^-$ concentrations on the surface of $MHCS_x$. **g** In-situ Raman spectra of $MHCS_{0.5}$ electrode at open circuit voltage and applied potential (0.1 V vs RHE) in 0.1 M PBS. **h** Comparison of in-situ Raman spectra on $MHCS_x$ electrode at applied potential (0.1 V vs RHE).

production of $OH^-$ by-products. Additionally, it was observed that there is no linear relationship between the current densities and measured $OH^-$ concentrations for each carbon sphere electrode (Supplementary Fig. 49b, c). Rather, the electrode with the hollow-structured carbon sphere exhibited a significantly higher $OH^-$ concentration. Therefore, we attribute the localized $OH^-$ accumulation on the $MHCS_{0.5}$ electrode to the combined effects of its high electrochemical activity and favorable hollow geometric configuration. These results provide experimental evidence that supports the simulation results discussed above.

To investigate the effect of $O_2$ enrichment and local pH elevation on the $2e^-$ ORR within the carbon sphere-based MHNs, we utilized in-situ Raman spectroscopy to probe the catalyst-adsorbate interactions (*$O_2$, *OOH). In an oxygen atmosphere, the Raman spectra of a $MHCS_{0.5}$ electrode at the open-circuit voltage (OCV) and an applied potential of 0.1 V vs RHE are shown in Fig. 6g. The D band was fitted by the characteristic modes of D1, D2, D3, and D4 at ~1130, 1200, 1340, and 1500 cm$^{-1}$, respectively[74]. Compared with the OCV, the D peak widened and a D1 peak emerged, which reflects the adsorption of $O_2$ (*$O_2$), and a

corresponding sharper D4 peak indicates a greater formation of *OOH intermediate[52,76,77]. This phenomenon can be inferred as follows: the enrichment of $O_2$ and localized pH elevation in the MHNs accelerate the reaction rate, facilitating the activation of $O_2$, stabilizing *$O_2$ intermediates, and inhibiting the protonation process of *OOH intermediates (*OOH + H$^+$ + e$^-$ → *O + H$_2$O) associated with the $4e^-$ pathway, thereby promoting $H_2O_2$ generation[74,78]. As illustrated in Fig. 6h, the comparison of the in-situ Raman spectra of all the samples revealed the most pronounced O-O stretching vibrations ($v_{*O2}$ and $v_{*OOH}$) signals on the $MHCS_{0.5}$, indicating the accelerated reaction rate within the dominant conformation featuring favorable reaction microenvironments.

We present a summary on the functional attributes of the mesoporous hollow carbon sphere nanoreactor in the electrochemical $2e^-$ ORR. The mesoporous channels facilitate fluid acceleration to transport the generated $H_2O_2$ into the bulk solution, decreasing the likelihood of its further electro-reduction on the catalyst surface[45]. Besides, the accelerated electrolyte flow rapidly pumps substrates to enrich $O_2$. Additionally, the accelerated reaction rate and favorable

hollow configuration promote localized pH elevation, providing a suitable microenvironment for the 2e- ORR. FES, experimental results, and in-situ Raman spectroscopy verify these functions of the MHNs based on engineered carbon spheres, which can be compared to certain structural functions of cells: the mesoporous shell of the MHNs is analogous to the cell membrane and effectively controls the entry and exit of substrates and products; while the hollow interior of the MHNs, akin to the cytoplasm within cells, furnishes a conducive internal microenvironment for the reaction. Whilst our example may lack the intricate and sophisticated functional units found in biology, it nonetheless serves as a source of inspiration for the construction of nanoreactors through biomimicry.

## Discussion

In summary, this work explored the catalytic functionalities of carbon-based MHNs. In detail, we engineered the interiors of mesoporous carbon spheres, drawing inspiration from FES for fluid diffusion, thereby constructing a series of model nanoreactors. By utilizing electrochemical 2e- ORR as a probe reaction, we provide evidence of the enhanced diffusion, reactant enrichment, and microenvironmental modulation facilitated by the mesoporous hollow architectures. These catalytic functionalities are closely linked to the structural parameters of MHNs, emphasizing the importance of controlling the nanoreactor's microstructure for targeted reactions. With the catalytic functionalities of the nanoreactor, the optimized MHCS$_{0.5}$ catalysts demonstrate impressive activity and selectivity in H$_2$O$_2$ electrosynthesis under both alkaline and neutral conditions. Moreover, the generation of medical-grade H$_2$O$_2$ disinfectant in a flow cell device substantiates the significant practical potential of this nanoreactor engineering. Overall, this study showcases the successful implementation of nanoreactor engineering in screening catalysts with functional orientation, paving the way for a wide range of nanoreactor applications.

## Methods

### Synthesis of mesoporous carbon sphere nanoreactors with different internal structures

In a typical synthesis, a solution of 65 mL of ethanol and 15 mL of H$_2$O was mixed with 2.5 mL of NH$_3$·H$_2$O and 0.8 mL of formaldehyde at room temperature. Subsequently, 0.6 g of resorcinol and 6 mL of TPOS were added to the mixed solution under stirring. The reaction was stirred for 24 h at room temperature, and the products were obtained by centrifugation, followed by washing with water and ethanol, and drying in an oven. The as-obtained products were subjected to pyrolysis at a temperature of 800 °C for 2 h in N$_2$ with a heating rate of 2 °C min$^{-1}$, followed by removal of the silica template through etching with HF (5 wt%). The internal structure of the carbon spheres is determined by the temporal order of addition of resorcinol and TPOS. MHCS$_{0.1}$, MHCS$_{0.5}$, MHCS$_{0.7}$, and MCS$_{-0.5}$ were synthesized by adding resorcinol and TPOS at intervals of −20, 0, 20, and 80 min, respectively. CS were prepared without the addition of TPOS. The reduction of MHCS$_{0.5}$ was conducted in a tube furnace at 800 °C for 2 h under a mixed H$_2$/N$_2$ atmosphere, resulting in the product denoted as R-MHCS$_{0.5}$.

### Material characterizations

Scanning electron microscopy (SEM) images were acquired using a scanning microscope (SU8230, Japan). Transmission electron microscopy (TEM) was performed with Tecnai G2 F30 (America) and JEM-2100 (Japan). Powder X-ray diffraction data were obtained using a Rigaku D/Max2500PC diffractometer with Cu Kα radiation (λ = 1.5418 Å) across the 2θ range of 5–80°, with a scan speed of 5° min$^{-1}$ at room temperature. Raman measurements were conducted using a spectrometer (NanoWizard Ultra Speed & inVia Raman) with a laser having an excitation wavelength of 532 nm. Nitrogen sorption

isotherms were recorded at 77 K (Tristar 3020, USA). X-ray photoelectron spectroscopy (XPS) data were collected with a monochromatic Al source (KRATOS, Axis Ultra). Electron paramagnetic resonance (EPR) measurements were carried out using a Bruker ECS-EMX X-band EPR spectrometer at room temperature.

### Electrochemical measurements

All electrochemical measurements were conducted using an electrochemical workstation (PARSTAT 3000A-DX) with a three-electrode cell at room temperature. An Ag/AgCl electrode was used as reference electrode and equipped with a salt bridge to eliminate liquid boundary potentials and prevent cross-contamination between the solution of the study system and the reference electrode system. Additionally, a Pt sheet was employed as the counter electrode, and a catalyst-loaded rotating ring-disk electrode (RRDE) functioned as the working electrode. The RRDE assembly included a glassy carbon rotation disk electrode (disk area: 0.247 cm$^2$) and a Pt ring (ring area: 0.186 cm$^2$) with a collection efficiency (N) of 0.37.

In the preparation of the catalyst ink, 5.0 mg of the electrocatalyst was dispersed in 950 μL of ethanol and 50 μL of a 5 wt% Nafion solution. After ultrasonic treatment for 30 min, 5.0 μL of the catalyst ink was drop-cast onto the disk electrode.

The electrochemical measurements were conducted at room temperature in O$_2$-saturated 0.1 M KOH aqueous solution (pH 13) and 0.1 M PBS (pH 7). The working electrode rotated at a constant speed of 1600 rpm during the tests. All potentials were converted to reference the reversible hydrogen electrode (RHE) using the following equation:

$$E_{RHE} = E_{Ag/AgCl} + 0.197 + 0.059 \times pH \quad (1)$$

Stable linear sweep voltammetry (LSV) was performed at a scan rate of 10 mV s$^{-1}$ in electrolytes saturated with N$_2$ and O$_2$. To ensure the complete oxidation of H$_2$O$_2$, the ring electrode was maintained at a constant potential of 1.2 V vs RHE. The selectivity of H$_2$O$_2$ and the electron transfer number (n) were calculated were calculated based on the disk current ($I_d$) and ring current ($I_r$) as following equations:

$$H_2O_2\% = \frac{200 \times I_r}{(N \times |I_d|) + I_r} \quad (2)$$

$$n = \frac{4 \times I_d}{|I_d| + I_r/N} \quad (3)$$

### Measurement of electrochemically active surface area

The electrochemically active surface area (ECSA) was determined using the double-layer capacitance method. Constant potential cyclic voltammetry (CV) scans were performed in the double-layer region, ranging from −0.06 to 0.04 V vs Ag/AgCl, with scan rates of 20, 40, 60, 80, and 100 mV s$^{-1}$. Plotting the sweep speed as the abscissa and ($j_a − j_c$) as the ordinate, the slope value was calculated to represent the double-layer capacitance ($C_{dl}$), which is linearly related to ECSA. ECSA was determined by calculating $C_{dl}$ using the following equations:

$$ECSA = \frac{C_{dl}}{C_s} \times A_{GCE} \quad (4)$$

Where the specific capacitance ($C_s$) for the electrode surface, typically falling within the range of 20 to 60 μF cm$^{-2}$. Specifically, we adopted a value of 40 μF cm$^{-2}$. $A_{GCE}$ denotes the electrode surface area.

### Bulk electrolysis

The bulk electrosynthesis of H$_2$O$_2$ in 0.1 M KOH was conducted using a customized H-cell electrolyzer to further investigate the practical

performance of $H_2O_2$ electrosynthesis. The working electrode employed was a carbon-paper electrode ($1 \times 1\,cm^2$) coated with $MHCS_{0.5}$ ($0.25\,mg\,cm^{-2}$). As for the counter and reference electrodes, a Pt electrode and an Ag/AgCl (saturated KCl) electrode were utilized, respectively. Chambers were separated by a Nafion membrane. The electrolyte consisted of 30 mL $O_2$-saturated 0.1 M KOH aqueous solution.

### Direct detection of the local pH changes on electrode

For the indirect observation of changes in local pH, the $MHCS_{0.5}$ catalyst was employed to facilitate the $2e^-$ ORR process for 10 min in the electrolytic cell (0.1 M PBS, pH = 7, 0.4 V vs RHE). Subsequently, the applied voltage was switched off, and the working electrode was transferred to 5 mL of aqueous solution and rinsed. After repeating these two steps 10 times, the pH value was measured in the reagent bottle using a pH meter. To ensure experimental accuracy, this parallel experiment was conducted 5 times.

For a direct observation of pH elevation on the electrode surface, we monitored the color change on the $MHCS_{0.5}$ electrode surface during the $2e^-$ ORR at 0.1 V vs RHE. The reaction was conducted in a H-type electrolytic cell using 0.1 M $K_2SO_4$ containing 1 mg of phenolphthalein as the electrolyte for visualization.

### $H_2O_2$ yield in the flow cell

The electrochemical synthesis of $H_2O_2$ via ORR was carried out using a two-compartment flow cell setup separated by a Nafion membrane. The cathode assembly in the gas diffusion layer was created by applying the $MHCS_{0.5}$ catalyst ink onto the carbon paper, with an actual working area of 1 $cm^2$ and a catalyst loading of $0.25\,mg\,cm^{-2}$. Ag/AgCl was employed as the reference electrode, and $IrO_2$ served as the anode. In the flow cell test, a 0.1 M KOH/0.1 M PBS electrolyte (40 mL) was circulated through each compartment at a flow rate of 6 mL $min^{-1}$, while the cathode received a continuous supply of $O_2$ at a rate of 20 mL $min^{-1}$.

### $H_2O_2$ concentration measurement

The amount of $H_2O_2$ was determined using a conventional cerium sulfate titration method, following the equation:

$$2Ce^{4+} + H_2O_2 = 2Ce^{3+} + 2H^+ + O_2 \qquad (5)$$

In this method, yellow $Ce^{4+}$ is reduced by $H_2O_2$ to colorless $Ce^{3+}$. A 0.5 mmol $L^{-1}$ $Ce(SO_4)_2$ solution was typically prepared by dissolving 16.6 mg $Ce(SO_4)_2$ in 100 mL of 0.50 mol $L^{-1}$ $H_2SO_4$ acid solution. The calibration curve was established by linear fitting of absorbance values at a wavelength of 320 nm for known concentrations of 0.01, 0.02, 0.05, 0.1, 0.2, 0.3, 0.4, and 0.5 mmol $L^{-1}$ $Ce^{4+}$. Using this calibration curve, the concentration of reduced $Ce^{4+}$ in the sample solution was determined by measuring UV–vis absorption intensity (Supplementary Fig. 50). Finally, the $H_2O_2$ yield was calculated based on the known concentration of reduced $Ce^{4+}$. The faradaic efficiency (FE) for $H_2O_2$ generation is calculated as the following equation:

$$FE\,(\%) = \frac{100\% \times \text{mole of generated } H_2O_2 \times 2 \times 96485}{\text{total consumed charge}} \qquad (6)$$

### Finite element simulation

A simulation was conducted using COMSOL Multiphysics software to investigate the relationship between electrolyte flow and mass transport in the mesoporous hollow nanoreactor (MHN) model, accounting for the convective and diffusive effects of $O_2$, $H_2O$, and $OH^-$. Both 3D and 2D models were employed to simulate the flow rate and material distribution in the MHNs, respectively.

To model the synthesized $MHCS_x$, MHNs were constructed uniformly with a particle radius of 150 nm and a mesopore channel of approximately 20 nm, with the proportion of the hollow component as the sole variable. This approach effectively illustrates the impact of various $d/r$ ratios on fluid transport at a standardized particle size, optimizing computational efficiency without compromising accuracy. Additionally, models with different pore sizes were simulated and compared. For simplicity, deformation was disregarded, and the model was assumed to be in a fixed position. Two 10 nm spaces were established on the left and right sides of the mesoporous channel to simulate the inflow and outflow of substances in the nonporous structure. The simulation model employed axisymmetric geometry to calculate fluid velocity and pressure in the mesoporous and hollow regions, as well as the concentration distribution of reactant substances. In this model, the left and right sides were designated as the inlet and outlet of the electrolyte, respectively, with mass transfer driven by fluid flow and concentration diffusion. The governing equations are as follows:

$$\rho(\mathbf{u} \cdot \nabla)\mathbf{u} = \nabla \cdot [-p\mathbf{I} + \mathbf{K}] + \mathbf{F} \qquad (7)$$

$$\rho\nabla \cdot \mathbf{u} = 0 \qquad (8)$$

$$\mathbf{K} = \mu(\nabla\mathbf{u} + (\nabla\mathbf{u})^T) \qquad (9)$$

$$\nabla \cdot \mathbf{J}_i + \mathbf{u} \cdot \nabla c_i = R_i \qquad (10)$$

$$\mathbf{J}_i = -D_i\nabla c_i \qquad (11)$$

where the dependent variables are velocity ($\mathbf{u}$), pressure ($p$), and concentration ($c_i$). $\rho$ is the fluid density, $\mu$ is the fluid viscosity, $D_i$ is the diffusion coefficient of substance i, and $R_i$ is the reaction rate. For the 3D/2D model of MHNs, electrolyte ingress occurs from the left, traverses the model region, and egresses from the right. To emulate the flow induced by solution rotation, the inlet velocity was set to 0.235 m $s^{-1}$, with a zero-pressure boundary condition at the outlet. Given the low Reynolds number regime (Re ~ $10^{-4}$) resulting from the small size of $MHCS_x$, the "Laminar Flow" module was chosen to simulate the flow field. At the model's inlet boundary, a solution saturated with $O_2$ was set, having a concentration of 21.4 mg $L^{-1}$ (0.66875 mol $m^3$) at the top inflow boundary. The outflow boundary was configured with a diffusion flux set to 0. The model's surface was designated as the location where the reaction occurs, and $O_2$ is consumed for conversion. Throughout the simulation, we examined the impacts of geometric dimensions, such as mesopore diameter and spacing, on $O_2$ and $H_2O_2$ concentrations. The geometric parameters required for the simulation were derived from experimental data, including electron microscope images.

## Data availability

The main data supporting the findings of this study are available within the main text and the Supplementary Information file. The main data generated in this study have been deposited in the figshare database under the accession code https://doi.org/10.6084/m9.figshare.24935178.

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

## Acknowledgements

This work was financially supported by the National Natural Science Foundation of China (No. 52172217, to J.Y.), China Postdoctoral Science Foundation (No. 2023M742391, to L.J.), Natural Science Foundation of Guangdong Province (No. 2021A1515010144, to J.Y.), Shenzhen Science and Technology Program (No. JCYJ20210324120400002 and ZDSYS20220527171401003, to J.Y.).

## Author contributions

Qiang Tian: Methodology, Investigation, Data Curation, Visualization, Writing-Original Draft, Writing-Review & Editing. Lingyan Jing: Funding Acquisition, Investigation, Data Curation, Suggestions and Comments, Writing-Reviewing & Editing. Hongnan Du, Yunchao Yin, Xiaolei Cheng, Jiaxin Xu, and Junyu Chen: Partial Exploration. Zhuoxin Liu: Suggestions and Comments, Writing-Reviewing & Editing. Jiayu Wan: Writing-Reviewing & Editing. Jian Liu: Suggestions and Comments, Writing-Reviewing & Editing. Jinlong Yang: Funding Acquisition, Supervision, Suggestions and Comments, Writing-Reviewing & Editing.

## Competing interests

The authors declare no competing interests.
