## [Peer Review File · Nature Communications]

REVIEWER COMMENTS

Reviewer #1 (Remarks to the Author):

It is worthy of high compliment that the authors introduced a novel nanoreactor engineering by employing engineered mesoporous carbon spheres as nanoreactors for electrocatalytic 2e- ORR processes. Leveraging the diffusion-enhanced and microenvironment-regulated capabilities of these nanoreactors, the authors achieved highly efficient electrosynthesis of H₂O₂. Furthermore, they have successfully demonstrated the production of medical-grade H₂O₂ disinfectant using a flow cell device, highlighting its remarkable catalytic efficiency and practicality. Therefore, I recommend the acceptance of this manuscript by Nature Communications after addressing the following concerns.

1. The authors have demonstrated the catalytic functionalities linked to the geometric parameters of the carbon spheres. It would be valuable to investigate whether the catalyst's structure and composition remains intact during electrolysis, and this can be further assessed through straightforward characterization techniques, such as TEM, XRD, and XPS.
2. Electronic conductivity can potentially impact electrocatalytic performance. Even though the carbon spheres in this study originated from the same carbon source, it remains important to ascertain this through electrochemical impedance spectroscopy.
3. MHCS0.5, which shows the best 2e-ORR activity among prepared samples, has an average particle radius of 210 nm and a hollow radius of 105 nm. However, when conducted finite element simulation, the model of MHCS0.5 consisting of a spherical shell with a radius of 150 nm, which is more closed to the MHCS0.7 and MHCS0.1. Please explain it a little bit more.
4. ECSA can assess the intrinsic active sites of the catalyst or the accessibility of these active sites. The authors determined ECSA using double-layer capacitance. To illustrate the relationship between the catalyst's active sites and its catalytic function, it is advisable for the authors to examine ECSA and consider normalizing electrochemical activity or H₂O₂ partial current.
5. When employing the RRDE method to assess catalyst stability over an extended duration, the accumulation of H₂O₂ can impact the ring current. However, the curves in this study exhibit a consistent ring current. How did the authors address the issue of H₂O₂ accumulation?

6. The production of H₂O₂ disinfectant under neutral or acidic conditions is more practical, and this can be emphasized in the manuscript.

7. There are many related catalysts with better performance in other reported works, which are not demonstrated in the Tables S3-S5. The authors should add them in the Tables.

Reviewer #2 (Remarks to the Author):

In this paper, the authors demonstrated the mass transfer diffusion in the mesoporous hollow sphere model using FES methods in fluid fields, followed by experimental applications of this hollow structure in catalysis. Several mesoporous hollow carbon sphere nanomaterials were synthesized using SiO₂ as a template with varied experimental sequences and timing. These were characterized using TEM, SEM, STEM with EDS analysis, BET surface analysis, XRD analysis, Raman spectroscopy, and XPS spectroscopy. Their varied structure also exhibited different electrochemical performances for the 2-electron oxygen reduction reaction to generate hydrogen peroxide in both alkaline and near-neutral electrolytes, with over 16 h stability in flow cells. The reaction mechanism was also analyzed using in situ Raman spectroscopy and local pH tests. Overall, this paper chose a good model to analyze the effect of mesopores on the 2-electron oxygen reduction, and the electrochemical performance was satisfactory. Publication is suggested. However, the reviewer does have some questions regarding the model, electrochemical analysis, and the mechanism discussion, which need to be addressed.

Firstly, and crucially, the authors selected a mesoporous hollow sphere as the standard model, where the holes are cylindrical and isolated from each other. In actual scenarios, however, the pores in the materials are predominantly interconnected, slit-shaped pores (see TEM, SEM, and STEM images in Figure 2, and the Type H3 hysteresis loops in the N₂ adsorption and desorption isotherms in Figure 3). The reviewer questions whether the fluid field in these actual materials will mirror that in the models. For example, the fluid may flow through the interconnected channels on the surface without entering the hollow cores of the spheres.

Secondly, as mentioned, the N₂ adsorption and desorption isotherms in Figure 3a exhibited an H3 hysteresis loop, which demonstrated slit-shaped pores, not cylindrical pores as the authors indicated. In both the isotherms and the pore size distributions, no pores from the hollow cores were found. Additionally, the BET surface area increased while the average pore size decreased as the diameter of the hollow core expanded. Does this indicate the hollow core is not accessible or it is made of a set of smaller pores? If that's the case, the model might require adjustments. Furthermore, the N₂ adsorption

decreased when the P/P0 value rose from 0.4 to 0.8 in Supplementary Figure 22, which does not make sense for N2 adsorption.

Thirdly, in Figure 4a, the onset potential varied for different MHCSx materials. As per the given assumption, the only difference among MHCSx materials is the pore size and thus, the mass transfer. Yet, onset potential is predominantly dictated by the intrinsic catalytic properties of the active sites. The reviewer wonders whether the active sites across the different MHCSx materials are consistent and if the divergence in electrochemical performance stems from differences in active sites rather than solely from mass transfer. The variability of active sites in mesoporous carbon materials, in comparison to microporous materials, was addressed by Bao, et al. in ACS Sustainable Chem. Eng. 2018, 6, 1, 311–317. In Figure 4d, MHCSx exhibited varied Tafel slopes, which are unaffected by the mass transfer process. This suggests that the reaction mechanism differs across the MHCSx samples.

Fourthly, Figure 4g intriguingly reveals that H₂O₂ selectivity consistently rose as electrolysis progressed within the same electrolyte. What's the reason for this phenomenon? Typically, the ring current either remains constant or decreases due to pollutant adsorption on the Pt ring, necessitating cleaning of the Pt ring every few hours. If the ring isn't contaminated, why was there a need to replace the electrolyte in the experiment?

Fifthly, in Figure 4i, Figure 4i showcases the disc current LSV of MHCSx materials in 0.1 M PBS, presenting two sets of limiting currents with distinct slopes. The reviewer questions if this indicates the presence of varied reaction mechanisms at different potentials.

Sixthly, based on the experimental design, variations in mass transport shouldn't impact the electrochemically active surface area, which could influence the active site concentration and limiting current. What caused this?

Minors:

Ag/AgCl is not a stable reference in alkaline solutions due to the formation of Ag₂O, which alters the reference potential.

Reviewer #3 (Remarks to the Author):

This study investigates the effect of geometric and porous structures of mesoporous hollow carbon nanospheres (MHCS) on two-electron oxygen reduction reaction ($2e^-$ ORR) activity for H_2O_2 electrosynthesis. The authors conducted fluid dynamics simulations to prove the advantages of mesoporous hollow structures, which could be attributed to facile O_2 supply as well as H_2O_2 emission. A set of MHCSs was prepared with carefully controlled structural properties, yet with similar surface oxygen functional groups, thereby enabling model catalyst study. It is demonstrated that MHCS0.5 exhibits the highest H_2O_2 electrosynthesis activity owing to its optimal mesostructures which enables fast diffusion. The structure-dependent diffusion behavior was tried to be proved experimentally, where however, some issues should be addressed during the revision.

Overall, this reviewer would like to recommend the publication in Nature Communications, but after major revision. Detailed comments are listed below.

1. The density of mesopore is presumed to have a strong effect on the fluid behavior. This effect can be shown and briefly discussed in the revised manuscript.
2. The mismatch between the BET surface area and double layer capacitance trends of MHNs also suggests the effective diffusion of MHCS0.5 compared to other MHCSs because high double layer capacitance requires sufficiently fast supply of electrolyte ions. The authors can add some discussion on the relation between them.
3. It is wondered whether the H_2O_2 selectivity difference between MHCS0.5 (the best) and MHCS-0.5 (the worst) is demonstrated in flow cell experiments.
4. The last experimental results where the MHCS0.5-based electrode contained the largest amounts of OH^- ions (the side products of ORR in neutral solutions) among the samples are interpreted that MHCS0.5 catalyst had a poorer ability of exhausting products than the other samples, which is contradictory to the authors' claim. This reviewer guesses that this is just a result of faster electrocatalytic reaction with MHCS0.5 as a catalyst than the others. The authors should explain this and provide a time-current figure of the catalysts obtained in "Direct detection of the local pH changes on electrode" experiments.
5. Similarly, the in-situ Raman results shown in Fig. 6h, where MHCS0.5 exhibited the highest signals for $*O_2$ and $*OOH$ among the samples, were consequences of higher activity of MHCS0.5 than the other samples. Therefore, the authors should reinvestigate the results and the related sentences need tone-down.

Responses to Reviewers' Comments

Reviewer #1

General comment:

It is worthy of high compliment that the authors introduced a novel nanoreactor engineering by employing engineered mesoporous carbon spheres as nanoreactors for electrocatalytic $2e^-$ ORR processes. Leveraging the diffusion-enhanced and microenvironment-regulated capabilities of these nanoreactors, the authors achieved highly efficient electrosynthesis of H_2O_2 . Furthermore, they have successfully demonstrated the production of medical-grade H_2O_2 disinfectant using a flow cell device, highlighting its remarkable catalytic efficiency and practicality. Therefore, I recommend the acceptance of this manuscript by Nature Communications after addressing the following concerns.

Response: We would like to thank Reviewer #1 for the valuable comments and positive recommendation. In response to the concerns raised by the reviewer, we have included additional experimental data and conducted thorough point-to-point revisions.

Comment 1-1:

The authors have demonstrated the catalytic functionalities linked to the geometric parameters of the carbon spheres. It would be valuable to investigate whether the catalyst's structure and composition remains intact during electrolysis, and this can be further assessed through straightforward characterization techniques, such as TEM, XRD, and XPS.

Response: We appreciate the reviewer's suggestion. As a response, we have included supplementary TEM images, XRD patterns, and XPS survey spectra for MHCS_{0.5} after electrolysis (Supplementary Fig. 37-38). These visual characterizations confirm the structural and compositional stability of MHCS_{0.5} during electrolysis. Additionally, we have integrated a relevant discussion into the revised manuscript, highlighted by yellow.

Supplementary Fig. 37. a, TEM image and **b**, XRD patterns of MHCS_{0.5} after long-term electrolysis.

Supplementary Fig. 38. a, XPS survey spectra of MHCS_{0.5} after long-term electrolysis. **b, c**, High-resolution XPS spectra of C 1s and O 1s of MHCS_{0.5} after long-term electrolysis.

(Page 10, Line 8-9) “After long-term electrolysis, MHCS_{0.5} still maintains its hollow mesoporous structure, and its chemical composition remains consistent with its pre-reaction state, ensuring its stability and reusability (Supplementary Fig. 37-38).”

Comment 1-2:

Electronic conductivity can potentially impact electrocatalytic performance. Even though the carbon spheres in this study originated from the same carbon source, it remains important to ascertain this through electrochemical impedance spectroscopy.

Response: We agree with the reviewer’s suggestion and have included electrochemical impedance spectroscopy (EIS) test to address this concern (Fig. R1). The EIS measurements were conducted in the three-electrode cell by applying a 10 mV alternating signal relative to the reference electrode across the frequency range of 0.1 Hz to 1000 kHz. Similar curve configurations indicate consistent electronic conductivity on MHCS_x, suggesting that varied geometrical parameters on MHCS_x do not impact electron conduction on their carbon matrix.

Fig. R1. Electrochemical impedance spectra (EIS) curves of MHCS_x.

Comment 1-3:

MHCS_{0.5}, which shows the best 2e⁻ ORR activity among prepared samples, has an average particle radius of 210 nm and a hollow radius of 105 nm. However, when conducted finite element simulation, the model of MHCS_{0.5} consisting of a spherical shell with a radius of 150 nm, which is more closed to the MHCS_{0.7} and MHCS_{0.1}. Please explain it a little bit more.

Response: We acknowledge the reviewer’s suggestion. As shown in Supplementary Table 1, the particle radius distribution of MHCS_x ranges between 150-210 nm. To simplify the model and reduce computational effort, we uniformly utilized the 150 nm model for constructing hollow mesoporous nanoreactors with varying hollow ratios, without compromising the accuracy of the predicted outcomes. As a model of the predictive mechanism, it is best to control only for variations in the degree of hollowing as the sole variable for elucidating the mechanism. Therefore, we have standardized all model radius to 150 nm.

To address this concern raised by the reviewer, we have included an explanation in both the “Finite element simulation” section and the main text (highlighted by yellow).

(Page 15, Line 27-30) “To simulate the synthesized MHCS_x, MHNs were uniformly constructed using a particle radius of 150 nm and a mesopore channel of about 20 nm, with the proportion of the hollow as the sole variable. In this way, the impact of various *d/r* on fluid transport is effectively demonstrated at a standardized particle size, optimizing computational power usage without compromising the accuracy of the findings.”

(Page 3, Line 9-11) “To ensure systematic construction of the MHNs and conserve computational resources for investigating the regularity of the electrolyte fluid, we standardized the particle size to 300 nm while maintaining the principle of unique variability for different d/r or ϕ .”

Comment 1-4:

ECSA can assess the intrinsic active sites of the catalyst or the accessibility of these active sites. The authors determined ECSA using double-layer capacitance. To illustrate the relationship between the catalyst's active sites and its catalytic function, it is advisable for the authors to examine ECSA and consider normalizing electrochemical activity or H_2O_2 partial current.

Response: This suggestion from the reviewer is highly insightful. To address this concern, we standardized the electrochemical activity and H_2O_2 partial currents based on the electrochemically active area in the revised manuscript (Fig. 4e). As illustrated in Fig. 4e, the normalized electrochemical activity, H_2O_2 partial current, and H_2O_2 selectivity at diffusion-dependent potentials show a positive correlation with the simulated inflow rate. Excluding the effects on accessibility for active site, it is evident that diffusive mass transfer contributes to the enhancement of ORR activity and the promotion of two-electron selectivity. We have updated the figure and adjusted the description accordingly in the revised manuscript (highlighted by yellow).

Fig. 4e. Normalized electrochemical activity and H_2O_2 partial current on $MHCS_x$ (@ 0.6 V vs RHE), along with the corresponding H_2O_2 selectivity and electrolyte inflow velocity.

(Page 9, Line 21-24) “Furthermore, we utilized ECSA to standardize the electrochemical activity and the H_2O_2 partial currents of $MHCS_x$ at diffusion-related potentials (0.6 V vs RHE), as illustrated in Fig. 4e. Interestingly, the associated normalized currents and $2e^-$ selectivity exhibit a positive correlation with the simulated electrolyte inflow velocity, providing additional evidence for the connection between the diffusion effect and the electrochemical performance on these MHNs.”

(Page 14, Line 24-28) “The ECSA was determined through the calculation of the C_{dl} using the following formula:

$$ECSA = \frac{C_{dl}}{C_s} \times A_{GCE}$$

Where the specific capacitance (C_s) of the electrode surface typically ranges from 20 to 60 $\mu\text{F cm}^{-2}$, we adopted 40 $\mu\text{F cm}^{-2}$ in this work. A_{GCE} refers to the electrode surface area.”

Comment 1-5:

When employing the RRDE method to assess catalyst stability over an extended duration, the accumulation of H_2O_2 can impact the ring current. However, the curves in this study exhibit a consistent ring current. How did the authors address the issue of H_2O_2 accumulation?

Response: In continuous testing, the gradual accumulation of H_2O_2 leads to a slight increase in the ring current, resulting in a slight increase in selectivity. To maintain the consistency of the ring current, we changed the electrolyte every 3 hours and rapidly performed electrochemical scavenging of the Pt ring to remove the impact of the accumulated H_2O_2 during continuous operation. We have further discussed and added the corresponding explanation in the revised manuscript (highlighted by yellow).

(Page 10, Line 2-4) “During the continuous testing, the accumulated H_2O_2 might lead to a slight rise in the ring current and subsequently in selectivity. To maintain a consistent ring current, we replaced the electrolyte every 3 h and performed electrochemical cleaning of the Pt ring to remove any effects of the accumulated H_2O_2 during continuous operation.”

Comment 1-6:

The production of H_2O_2 disinfectant under neutral or acidic conditions is more practical, and this can be emphasized in the manuscript.

Response: We greatly appreciate this suggestion from the reviewer. In response, we have included a discussion on the practicality of producing H_2O_2 disinfectants in neutral condition in the revised manuscript (highlighted by yellow).

(Page 10, Line 38-41) “Notably, the electroproduction of H_2O_2 solutions in neutral electrolytes is particularly appealing due to its environmental friendliness, minimal corrosiveness, and potential for

reduced electrolytic cell costs⁷². Additionally, neutral H₂O₂ solutions provide a sustainable and versatile option for practical applications, including the direct use of synthesized H₂O₂ in biochemical systems^{73, 74}.”

(Page 11, Line 7-8) “Significantly, the medical-grade H₂O₂ solution produced under neutral conditions can be elegantly stored or directly employed for disinfection and sterilization purposes.”

Comment 1-7:

There are many related catalysts with better performance in other reported works, which are not demonstrated in the Tables S3-S5. The authors should add them in the Tables.

Response: In response to this concern raised by the reviewer, we have included recently reported relevant electrocatalysts in Supplementary Table S3-S5. The tables and the literature added are listed below, highlighted by yellow.

Supplementary Table. 3. Comparison of electrocatalytic H₂O₂ production performance by RRDE technique in 0.1 M KOH electrolyte (pH = 13) of recently reported electrocatalysts.

Catalyst	Onset potential (V _{RHE})	H ₂ O ₂ selectivity (%)	Potential (V _{RHE})	Ref.
MHCS _{0.5}	0.85	>95	0.3-0.8	This work
Co-N/HPC	0.8	~95	0.25-0.65	1
CQD	0.8	>90 >95	0.25-0.65 0.45-0.65	2
O-HGr	0.78	>90 ~95	0.2-0.75 0.45-0.75	3
CF	0.81	~90	0.55-0.63	4
N-FLG-8	0.76	>95	0.35-0.7	5
OCNS ₈₀₀	0.82	~90	0.5-0.75	6
NiB ₂	0.68	>90 ~95	0.2-0.6 0.4-0.6	7
O-GOMC	0.8	~90	0.2-0.75	8
Mo ₁ /OSG-H	0.78	~95	0.4-0.7	9

Co-N-C	0.78	70-80	0.1-0.8	10
Ni-N ₂ O ₂ /C	0.7	>90 ~95	0.1-0.5 0.4-0.5	11
Cu-Pb	0.7	>90 ~95	0.4-0.57 0.52-0.57	12
Fe-CNT	0.82	~90	0.55-0.75	13
ZnO ₃ C	0.73	~80	0.5-0.7	14
Co-POC-O	0.82	~80	0.55-0.8	15
Pd@Au _{0.95} Pd _{0.05}	0.7	~90	0.4-0.7	16
Bi ₂ Te ₃	0.76	>95	0.2-0.6	17
GLC	0.8	~90	0.4-0.6	18
CoNPs@N/C	0.83	~90	0.1-0.6	19
OCG-800	0.78	>95	0.3-0.7	20
CB-Plasma	0.8	>90 ~95	0.4-0.7 0.5-0.6	21
In SAs/NSBC	0.8	>90	0.4-0.7	22
P-NMG-10	0.78	80-90	0-0.7	23
N,S-TCNTs	0.78	~90	0.2-0.8	24
Fe _{sA} -NS/C-700	0.75	92	0.3-0.5	25
NBO-G/CNTs	0.8	80-90	0.25-0.75	26
FS-CFs	0.81	~85	0.5-0.8	27
In ₂ O ₃ /CDs-10	0.76	~95	0.4-0.7	28
Pb(NiWMnNbZrTi) _{1/6} O ₃	0.76	~95	0.3-0.7	29
Co-SCD-2	0.78	>90	0.1-0.6	30
NiPyPC/CN	0.77	~90	0.25-0.65	31

Supplementary Table. 4. Comparison of electrocatalytic H₂O₂ production performance by RRDE technique in neutral electrolyte (pH = 7) of recently reported electrocatalysts.

Catalyst	Electrolyte	Onset potential (V _{RHE})	j_{disk} (mA cm ⁻²) @ 0.2 V _{RHE}	Selectivity (%@V _{RHE})	Ref.
MHCS _{0.5}	0.1 M PBS	0.60	-2.8	85-90% @ (0.1-0.35) >90% @ (0.30-0.50)	This work
B-C	0.1 M Na ₂ SO ₄	0.45	-2.1	70-80% @ (0.1-0.35)	32
O-CNTs	0.1 M PBS	0.5	-0.24	80-90% @ (0.2-0.5)	33
MBC-2	0.5 M Na ₂ SO ₄	0.4	-1.6	80-90% @ (0.2-0.4)	34
N-FLG-8	0.1 M Na ₂ SO ₄	0.5	-2.5	70-80% @ (0.1-0.40)	5
Co-N-C	0.1 M PBS	0.62	-2.5	>60% @ (0.3-0.4)	10
h-SnO ₂	0.1 M Na ₂ SO ₄	0.45	-2.1	94-99% @ (0-0.40)	35
CoPc-CNT(O)	0.1 M K ₂ SO ₄	0.52	-2.7	>90% @ (0.35-0.55)	36
α -PdSe ₂ NPs/C	0.1 M Na ₂ SO ₄	0.4	-1.8	>90% @ (0-0.30)	37
ZnNP-O-C	0.1 M PBS	0.57	-0.2	> 90% @ (0-0.40)	38
Fe-CNT	0.1 M PBS	0.53	-2.8	> 90% @ (0.30-0.50)	13
MCNS	0.1 M PBS	0.53	-1.8	> 90% @ (0.13-0.43)	29
O-C(Al)	0.1 M PBS	0.52	-2.3	85-90% @ (0.1-0.45)	30
L-ZnO	0.6 M K ₂ SO ₄	0.38	-0.9	~ 90% @ 0.2	39
ZnO@ZnO ₂	0.1 M K ₂ SO ₄	0.42	-2.7	> 90% @ (0-0.4)	40
In SAs/NBSC	0.1 M PBS	0.5	-2.7	> 90% @ (0.1-0.35)	22

ZnCo-ZIF	0.1 M PBS	0.48	-1.8	~ 90 @ (0.05-0.3)	41
O-HGr	0.1 M PBS	0.5	-1.9	~ 91 @ 0.2	3

Supplementary Table. 5. Comparison of electrochemical performance in the flow cell device with recently reported electrocatalysts.

Catalysts	Electrolyte	Condition	H ₂ O ₂ yield (mol g ⁻¹ h ⁻¹)	FE (%)	Ref.
MHCS _{0.5}	0.1 M KOH	0.1 V vs RHE	17.18	90	This work
MHCS _{0.5}	0.1 M PBS	0.1 V vs RHE	12.64	90	This work
FeSA-NS/C-700	0.1 M KOH	0 V vs RHE	4.95	91.4	25
CNB-ZIL 8	0.1 M KOH	-1.4 V (cell voltage)	1.79	80	42
Ni-N ₂ O ₂	0.1 M KOH	70 mA cm ⁻²	5.9	90	11
N-FLG-8	1 M KOH	1.8 V (cell voltage)	9.66	90	5
Co-NC	0.1 M KOH	50 mA cm ⁻²	4.2	42	43
N-O-C-800	0.1 M KOH	0.25 V vs RHE	1.47	65	44
OCNS900	0.1 M KOH	0.2 V vs RHE	0.77	60	6
ZnCo-ZIF	0.1 M KOH	60 mA cm ⁻²	4.3	70	41
ZnCo-ZIF	0.1 M PBS	70 mA cm ⁻²	3.8	72	41
Ni ₄ -B ₁ @BNC	1 M KOH	0.2 V vs RHE	0.13	73	45
Sb-NSCF	1 M KOH	0.55 V vs RHE	7.46	80	46
In ₂ O ₃ /CDs-10	0.1 M KOH	0.5 V vs RHE	4.5	93	28
CoPc-OCNT	1 M KOH	300 mA cm ⁻²	11.527	96	39
h-SnO ₂	1 M KOH	0.5 V vs RHE	3.18	/	35
h-SnO ₂	1.0 M Na ₂ SO ₄	0 V vs RHE	3.8	72	35
ZnO@ZnO ₂	0.1 M K ₂ SO ₄	0.1 V vs RHE	5.47	95	40

In SAs/NSBC	0.1 M Na ₂ SO ₄	90 mA cm ⁻²	6.71	75	22
In SAs/NSBC	0.1 M KOH	90 mA cm ⁻²	6.49	78	22
Co-N-C	0.5 M NaCl	50 mA cm ⁻²	4.5	75	47
NBO-G/CNTs	0.1 M KOH	50 mA cm ⁻²	0.71	81	26
PBT	1.0 M KOH	100 mA cm ⁻²	3.13	96	48
PANI/CDs-Co-2	0.1 M KOH	0 V vs RHE	3.5	86	49
NiB ₂	0.1 M KOH	0.4 V vs RHE	4.75	90	7

Supplementary References

“3. Koh KH, Bagherzadeh Mostaghimi AH, Chang Q, Kim YJ, Siahrostami S, Han TH, *et al.* Elucidation and modulation of active sites in holey graphene electrocatalysts for H₂O₂ production. *EcoMat* **5**, e12266 (2023).”

“7. Wu J, Hou M, Chen Z, Hao W, Pan X, Yang H, *et al.* Composition engineering of amorphous nickel boride nanoarchitectures enabling highly efficient electrosynthesis of hydrogen peroxide. *Adv. Mater.* **34**, 2202995 (2022).”

“22. Zhang E, *et al.* Engineering the local atomic environments of indium single-atom catalysts for efficient electrochemical production of hydrogen peroxide. *Angew. Chem. Int. Ed.* **134**, e202117347 (2022).

23. Peng W, *et al.* Facilitating two-electron oxygen reduction with pyrrolic nitrogen sites for electrochemical hydrogen peroxide production. *Nat. Commun.* **14**, 4430 (2023).

24. Long Y, *et al.* Tailoring the Atomic-Local Environment of Carbon Nanotube Tips for Selective H₂O₂ Electrosynthesis at High Current Densities. *Adv. Mater.* 2303905 (2023).

25. Li Y, *et al.* Single-atom Iron Catalyst with Biomimetic Active Center to Accelerate Proton Spillover for Medical-level Electrosynthesis of H₂O₂ Disinfectant. *Angew. Chem. Int. Ed.* e202306491 (2023).

26. Fan M, *et al.* N-B-OH Site-Activated Graphene Quantum Dots for Boosting Electrochemical Hydrogen Peroxide Production. *Adv. Mater.* 2209086 (2023).

27. Xiang F, *et al.* Enhanced Selectivity in the Electroproduction of H₂O₂ via F/S Dual-Doping in Metal-Free Nanofibers. *Adv. Mater.* 2208533 (2022).

28. Wu J, *et al.* The electron transport regulation in carbon dots/In₂O₃ electrocatalyst enable 100% selectivity for oxygen reduction to hydrogen peroxide. *Adv. Funct. Mater.* **32**, 2203647 (2022).

29. Chen Z, *et al.* Entropy enhanced perovskite oxide ceramic for efficient electrochemical reduction of oxygen to hydrogen peroxide. *Angew. Chem. Int. Ed.* **134**, e202200086 (2022).
30. Qi D, *et al.* Cyclodextrin-supported Co(OH)₂ Clusters as Electrocatalysts for Efficient and Selective H₂O₂ Synthesis. *Angew. Chem. Int. Ed.* **135**, e202307355 (2023).
31. Sun L, Jin X, Su T, Fisher AC, Wang X. Conjugated Nickel Phthalocyanine Derivatives for Heterogeneous Electrocatalytic H₂O₂ Synthesis. *Adv. Mater.* 2306336 (2023).”
- “39. Ding S, *et al.* An abnormal size effect enables ampere-level O₂ electroreduction to hydrogen peroxide in neutral electrolytes. *Energy Environ. Sci.* **16**, 3363-3372 (2023).
40. Zhou Y, *et al.* The operation active sites of O₂ reduction to H₂O₂ over ZnO. *Energy Environ. Sci.* **16**, 3526-3533 (2023).
41. Zhang C, *et al.* Crystal engineering enables cobalt-based metal–organic frameworks as high-performance electrocatalysts for H₂O₂ production. *J. Am. Chem. Soc.* **145**, 7791-7799 (2023).”
- “47. Zhao Q, *et al.* Approaching a high-rate and sustainable production of hydrogen peroxide: oxygen reduction on Co–N–C single-atom electrocatalysts in simulated seawater. *Energy Environ Sci.* **14**, 5444-5456 (2021).
48. Yang Z, Gao Y, Zuo L, Long C, Yang C, Zhang X. Tailoring Heteroatoms in Conjugated Microporous Polymers for Boosting Oxygen Electrochemical Reduction to Hydrogen Peroxide. *ACS Catal.* **13**, 4790-4798 (2023).
49. Zhou Y, *et al.* Efficient synthesis of H₂O₂ via oxygen reduction over PANI driven by kinetics regulation of carbon dots. *Appl. Catal. B* **322**, 122105 (2023).”

Reviewer #2

General comment:

In this paper, the authors demonstrated the mass transfer diffusion in the mesoporous hollow sphere model using FES methods in fluid fields, followed by experimental applications of this hollow structure in catalysis. Several mesoporous hollow carbon sphere nanomaterials were synthesized using SiO₂ as a template with varied experimental sequences and timing. These were characterized using TEM, SEM, STEM with EDS analysis, BET surface analysis, XRD analysis, Raman spectroscopy, and XPS spectroscopy. Their varied structure also exhibited different electrochemical performances for the 2-electron oxygen reduction reaction to generate hydrogen peroxide in both alkaline and near-neutral electrolytes, with over 16 h stability in flow cells. The reaction mechanism was also analyzed using in situ Raman spectroscopy and local pH tests. Overall, this paper chose a good model to analyze the effect of mesopores on the 2-electron oxygen reduction, and the electrochemical performance was satisfactory. Publication is suggested. However, the reviewer does have some questions regarding the model, electrochemical analysis, and the mechanism discussion, which need to be addressed.

Response: We are grateful to Reviewer #2 for the valuable comments and positive recommendation. We have incorporated additional experimental data and improved the textual descriptions to enhance the overall quality of the revised manuscript. Here, we have provided detailed responses and revisions addressing the concerns raised by the reviewer.

Comment 2-1:

Firstly, and crucially, the authors selected a mesoporous hollow sphere as the standard model, where the holes are cylindrical and isolated from each other. In actual scenarios, however, the pores in the materials are predominantly interconnected, slit-shaped pores (see TEM, SEM, and STEM images in Figure 2, and the Type H3 hysteresis loops in the N₂ adsorption and desorption isotherms in Figure 3). The reviewer questions whether the fluid field in these actual materials will mirror that in the models. For example, the fluid may flow through the interconnected channels on the surface without entering the hollow cores of the spheres.

Response: Thank you very much to the reviewer for the professional insights on our work. It is indeed accurate, as pointed out by the reviewer, that the mesoporous channels in our materials exhibit dendritic and interconnected features. However, it is important to note that a substantial portion of these mesopores traverse the shell layer, connecting the inner hollow and the outer structure, as evident from the SEM

images of the crushed MHCS_{0.5} sample and the locally magnified TEM image of MHCS_{0.5} (Supplementary Fig. 16).

As mentioned by the reviewer, we did not directly observe the flow of electrolyte through the hollow core in the original manuscript. To address the reviewer's query about whether the fluid flowed through the hollow, we conducted the following experiment: after the 2e⁻ ORR reaction using MHCS_{0.5} in 0.5 M KCl electrolyte and after 1 hour of electrolysis, we opted not to wash the MHCS_{0.5} material but instead dried it directly. Subsequently, samples were prepared for observation under TEM. As presented in Supplementary Fig. 17. a, b, some of the internal voids of the MHCS_{0.5} were filled or partially filled as a result of the electrolyte fluid flowing through the internal voids and retaining dry KCl. Additionally, we performed elemental face sweeps on the altered samples to demonstrate the distribution of elemental K in the filler (Supplementary Fig. 17. c-f). The corresponding constituent line sweep further confirm the distribution of K element in mesopores and hollows (Supplementary Fig. 18). From these observations, we can conclude that the shell layer of MHCS_{0.5} is distributed with mesoporous channels connecting the inside and outside, and the electrolyte fluid can indeed flow through the hollow space. We express our gratitude to the reviewer once more for this valuable suggestion, which is crucial for validating the simulation results and enhancing the quality of the work. To address the reviewer's concerns, we have added relevant data and explanatory notes to the revised manuscript (highlighted by yellow).

Supplementary Fig. 16. a, b, SEM images of crushed MHCS_{0.5} particles. c. Partially enlarged TEM image of MHCS_{0.5}.

Supplementary Fig. 17. a, b, TEM images of MHCS_{0.5} samples dried directly after 1-hour electrolysis in 0.5 M KCl electrolyte. c-f, HAADF-STEM image, and the corresponding elemental mapping images of MHCS_{0.5} dried directly after 1-hour electrolysis in 0.5 M KCl electrolyte.

Supplementary Fig. 18. a, Linear scan positions on MHCS_{0.5} sample dried directly after 1-hour electrolysis in 0.5 M KCl electrolyte, and b, the corresponding linear scans of C, O, and K elements.

(Page 4, Line 38-41) “Besides, some mesoporous channels running through the shell layer were clearly observed in both the SEM images of the crushed MHCS_{0.5} and the locally enlarged TEM images of MHCS_{0.5}, providing a path for the electrolyte fluid to pass through the internal hollow space (Supplementary Fig. 16).”

(Page 4, Line 42-44; Page 5, Line 1-4) “To confirm the flow of electrolyte through the internal hollow space of MHCS_{0.5}, we conducted an initial one-hour ORR using MHCS_{0.5} in 0.5 M KCl electrolyte. Following the reaction, the MHCS_{0.5} samples were dried without washing for direct TEM observation, revealing the presence of residual KCl components filling the interiors of MHCS_{0.5} (Supplementary Fig. 17). These findings are further validated by the corresponding elemental linear scans, confirming the presence of mesoporous channels running through the shell layers of MHCS_{0.5} and the flow of electrolyte fluids through these channels, enabling movement of electrolyte through the hollow interior (Supplementary Fig. 18).”

Comment 2-2:

Secondly, as mentioned, the N₂ adsorption and desorption isotherms in Figure 3a exhibited an H3 hysteresis loop, which demonstrated slit-shaped pores, not cylindrical pores as the authors indicated. In both the isotherms and the pore size distributions, no pores from the hollow cores were found. Additionally, the BET surface area increased while the average pore size decreased as the diameter of the hollow core expanded. Does this indicate the hollow core is not accessible or it is made of a set of smaller pores? If that's the case, the model might require adjustments. Furthermore, the N₂ adsorption decreased when the P/P₀ value rose from 0.4 to 0.8 in Supplementary Figure 22, which does not make sense for N₂ adsorption.

Response: We are deeply grateful to the reviewer for this valuable suggestion. In response to the **Comment 2-1**, we have demonstrated that, despite the existence of small slit pores in the carbon spheres, there are also mesoporous channels linking the internal hollow to the outside. Additionally, we have validated the flow of electrolyte fluids through the hollow space. And the corresponding manuscript revisions regarding this aspect have already been incorporated the revised manuscript.

We greatly appreciate the reviewer's observation that some of the mesopores within the carbon sphere are interconnected. In response, we attempted to modify the MHCS_{0.5} model to incorporate interconnected pores (Fig. R2a). Using the adjusted model, we conducted simulations of the flow velocity distribution, yielding comparable outcomes (Fig. R2b). The velocity distribution across the cross-section ($y = 0$) demonstrated the flow of fluid through the hollow region (Fig. R2c). Besides, we also conducted a comparison of the flow velocity distribution at specific locations $(-160, 0, 0 \rightarrow 160, 0, 0)$. We observed that, apart from a slight increase at the pore junction, the inflow and outflow rates of the electrolyte resemble those of the original model (Fig. R2d). Hence, we can deduce that the interconnected mesopores exert a negligible impact on the regularity of the flow velocity distribution.

We acknowledge the reviewer’s question regarding “the N_2 adsorption decreased when the P/P_0 value rose from 0.4 to 0.8 in Supplementary Figure 22, which does not make sense for N_2 adsorption”. In response to this concern, we have re-tested the nitrogen adsorption and desorption curve and the pore size distributions of the CS material. The updated results can be found in **Supplementary Fig. 25** in the revised manuscript, and the corresponding descriptions have been adjusted accordingly (highlighted by yellow).

Fig. R2. **a**, Simulation model of MHCS_{0.5} containing connected mesopore channels. **b**, Spatial distribution of simulated flow rates. **c**, Cross-section ($y = 0$) of the spatial flow velocity distribution. **d**, Comparison of flow rates for different models.

Supplementary Fig. 25. **a**, Nitrogen adsorption-desorption isotherms and the corresponding **b**, pore size distribution curves of CS.

(Page 7, Line 1-2) “Compared to MHCS_x, CS has a lower specific surface area of only 552 m² g⁻¹ and almost no mesopore distribution (Supplementary Fig. 25).”

Comment 2-3:

Thirdly, in Figure 4a, the onset potential varied for different MHCS_x materials. As per the given assumption, the only difference among MHCS_x materials is the pore size and thus, the mass transfer. Yet, onset potential is predominantly dictated by the intrinsic catalytic properties of the active sites. The reviewer wonders whether the active sites across the different MHCS_x materials are consistent and if the divergence in electrochemical performance stems from differences in active sites rather than solely from mass transfer. The variability of active sites in mesoporous carbon materials, in comparison to microporous materials, was addressed by Bao, et al. in *ACS Sustainable Chem. Eng.* 2018, 6, 1, 311–317. In Figure 4d, MHCS_x exhibited varied Tafel slopes, which are unaffected by the mass transfer process. This suggests that the reaction mechanism differs across the MHCS_x samples.

Response: We sincerely appreciate the reviewer's valuable suggestion. In response to the concerns raised, we delved deeper into the examination of the reactive active sites. Firstly, the reviewer provided a literature (*ACS Sustain. Chem. Eng.* 2018, 6, 1, 311-317) describing defects on mesoporous carbon as possible active sites for 2e⁻ ORR. To investigate defective sites on MHCS_x, we conducted electron paramagnetic resonance (EPR) tests. As indicated in Supplementary Fig. 30, the comparable signal strength from MHCS_x confirms their similar defect levels. Moreover, the comparable ratio of the D and G bands in the Raman spectra of MHCS_x implies a similar degree of carbon defects (Fig. 3d). These shared properties can be attributed to the common origin of MHCS_x, derived from resorcinol-formaldehyde resin-based carbon, and through the same carbonization process. In addition, we conducted a reduction of the MHCS_{0.5} material in a mixed H₂/N₂ atmosphere to eliminate O species. The resulting sample (R-MHCS_{0.5}) exhibited a reduced oxygen content of 1.76 at.%, leading to a significant decrease in its electrochemical activity and H₂O₂ selectivity (Supplementary Fig. 32). These findings indicate that the O species present on MHCS_x serve as the primary source of its 2e⁻ ORR activity, which aligns with previous reports (*Nat Catal.* 2018, 1, 156-162; *Nat Catal.* 2018, 1, 282-290; *Angew. Chem. Int. Ed.* 2021, 133, 16743-16750). It is worth noting that all MHCS_x samples are derived from the same carbon source and undergo similar preparation processes, resulting in comparable oxygen content and species (Fig. 3f in the revised manuscript). Therefore, these resemblances in active sites (defects and oxygen species) for 2e⁻ ORR were directly confirmed through examinations carried out on MHCS_x in an electrolyte-free environment.

However, it is important to consider the accessibility and utilization of intrinsic active sites in the electrolyte environment, which was our initial intent in designing the mesoporous hollow nanoreactors. The electrochemical performance of MHCS_x is subject to a combination of active site utilization and mass

transfer, a relationship intricately connected to the nanoreactor configuration. Intuitively, enhanced interaction area with electrolyte leads to high electrochemical activity; the frequency of contact with the catalyst surface, influenced by electrolyte flow, also impacts the extent of electrochemical activity; additionally, higher flow rates in the nanoreactor typically enhance $2e^-$ selectivity by facilitating the prompt transfer of the generated H_2O_2 . The reviewer raised the issue of different onset potentials on $MHCS_x$, which we speculate could be primarily attributed to the distinct utilization of active sites in the electrolyte by various nano-porous architectures. This phenomenon is commonly observed in porous carbon electrocatalytic materials (*Adv. Mater.* **2023**, 35, 5, 2208942; *J. Am. Chem. Soc.* **2019**, 141, 17, 7073-7080; *Nano Lett.* **2023**, 23, 11, 4699-4707; *Angew. Chem. Int. Ed.* **2018**, 57, 6176-6180). In our revised manuscript, we have bolstered the discussion on the active site and incorporated the literature suggested by the reviewer to enhance the elucidation (highlighted by yellow).

Supplementary Fig. 30. EPR spectra of $MHCS_x$.

Supplementary Fig. 32. **a**, XPS survey spectrum of R- $MHCS_{0.5}$. **b**, LSV curves of R- $MHCS_{0.5}$ recorded at 1600 rpm in an O_2 -saturated 0.1 M KOH and the corresponding H_2O_2 selectivity.

(Page 13, Line 42-44) “The electron paramagnetic resonance (EPR) measurements were conducted using a Bruker ECS-EMX X-band EPR spectrometer at the room temperature.”

(Page 7, Line 10-13) “In addition, defective carbon or some carbon edges, have been proposed as potential active sites for $2e^-$ ORR in some previous studies^{30, 54}. To examine the defective carbon, we utilized electron paramagnetic resonance (EPR) and observed nearly identical signal intensities on MHCS_x, suggesting that MHCS_x exhibits a consistent level of defectivity for electrochemical $2e^-$ ORR (Supplementary Fig. 30).”

(Page 7, Line 24-26) “The high ORR onset potential may stem from its appropriate porous structure and geometric framework, which enhance contact with electrolyte fluids to improve the utilization of carbon surface active sites in the electrolyte environment^{14, 19, 58}.”

(Page 8, Line 11-13; Page 9, Line 1-3) “We reduced the MHCS_{0.5} sample in a mixed H₂/N₂ atmosphere to eliminate the oxygen component, resulting in R-MHCS_{0.5} with an oxygen content as low as 1.76 at.% (Supplementary Fig. 32a). Consequently, the ORR activity and H₂O₂ selectivity of R-MHCS_{0.5} were significantly reduced (Supplementary Fig. 32b), indicating that the oxygen species on the carbon matrix serve as the active source of $2e^-$ ORR, consistent with some previous reports^{52, 59}. Given the comparable oxygen species on MHCS_x, the notable discrepancy in H₂O₂ selectivity is attributed to the fluid behavior of electrolytes within different architectures of MHNs.”

“54. Chen S, *et al.* Defective carbon-based materials for the electrochemical synthesis of hydrogen peroxide. *ACS Sustain. Chem. Eng.* **6**, 311-317 (2018).”

The Tafel slope (b) can usually be derived from the Butler-Volmer equation (*Electrochemical methods: fundamentals and applications*. John Wiley & Sons, 2022.): $\eta = \frac{RT}{\alpha F} \ln j_0 - \frac{RT}{\alpha F} \ln j = a - b \lg j$ (Tafel equation). It is important to note that the applicability of the Butler-Volmer equation is limited to situations where the electrochemical interface is in equilibrium, with equal positive and negative reaction rates and rate constants. Typically, small Tafel slopes in ORR tests are interpreted as indicative of fast kinetic processes (*Nat. Commun.* **2020**, *11*, 4173; *Science*, **2019**, *366*, 6467, 850-856). Besides, there is a “coverage” in the ORR that relies on the overpotential, making the existence of a “constant” reaction rate uncertain (*Scientific reports*, **2015**, *5*, 13801). To address the reviewer’s query regarding the Tafel slopes of MHCS_x and the reaction mechanism, we reviewed the relevant literatures (*J. Electrochem. Soc.* **2012**, *159*, H864; *J. Am. Chem. Soc.* **2021**, *143*, 7819-7827; *Angew. Chem. Int. Ed.* **2022**, *61*, e202213296),

speculated on the potential reaction rate-determining steps (RDS). For example, a Tafel slope of $\sim 120 \text{ mV dec}^{-1}$ corresponds to the first OOH* formation RDS ($\text{O}_2 + \text{H}^+/\text{e}^- + * \rightarrow \text{OOH}^*$), $\sim 70 \text{ mV dec}^{-1}$ corresponds to the H_2O_2 desorption RDS ($*\text{OOH} + \text{H}^+/\text{e}^- \rightarrow \text{H}_2\text{O}_2 + *$). During the literature review, we recognized that our initial delineation of the Tafel slopes, particularly very close to the onset potential, may lead to erroneous conclusions (*Cell Rep. Phys. Sci.* **2022**, *3*, 100987). Therefore, we re-delineated the Tafel slopes and modified the relevant description in the revised manuscript according to the suggestion raised by the reviewer (highlighted by yellow).

Additionally, we studied the ORR reaction mechanism on MHCS_x from the perspective of the H_2O_2 reduction reaction ($\text{H}_2\text{O}_2\text{RR}$). We have incorporated these discussions into the revised manuscript (highlighted by yellow).

(Page 9, Line 3-11) “Fig. 4d displays the Tafel slopes of MHCS_x , with $\text{MHCS}_{0.5}$ exhibiting a low value of 73 mV dec^{-1} , indicating its fast reaction kinetics. Tafel plots reflect various rate-determining steps (RDS), and a progressive decrease in Tafel slopes could indicate a shift in the RDS from the first electron-transfer step ($\text{O}_2 \rightarrow *\text{OOH}$, $\sim 120 \text{ mV dec}^{-1}$) to the last H_2O_2 desorption step ($*\text{OOH} \rightarrow \text{H}_2\text{O}_2$, $\sim 70 \text{ mV dec}^{-1}$)^{60, 61}. The ORR pathway was studied from the perspective of the H_2O_2 reduction reaction ($\text{H}_2\text{O}_2\text{RR}$) activity in N_2 -saturated 0.1 M KOH containing $10 \text{ mM H}_2\text{O}_2$. Among MHCS_x , $\text{MHCS}_{0.5}$ exhibits apparent inertness to $\text{H}_2\text{O}_2\text{RR}$, with negligible currents, confirming its highly selective 2e^- pathway (Supplementary Fig. 33). In contrast, we observed an increasing trend in $\text{H}_2\text{O}_2\text{RR}$ current from $\text{MHCS}_{0.7}$, $\text{MHCS}_{0.1}$ to $\text{MCS}_{0.5}$, suggesting an improved $\text{H}_2\text{O}_2\text{RR}$ capability. Considering the simulated fluid velocity on MHCS_x , we deduce that the low flow rates would advance the indirect 4e^- pathway to reduce the H_2O_2 selectivity^{61, 62}.”

Supplementary Fig. 33. LSV curves for $\text{H}_2\text{O}_2\text{RR}$ recorded on MHCS_x in N_2 -saturated 0.1 M KOH containing $10 \text{ mM H}_2\text{O}_2$.

Comment 2-4:

Fourthly, Figure 4g intriguingly reveals that H₂O₂ selectivity consistently rose as electrolysis progressed within the same electrolyte. What's the reason for this phenomenon? Typically, the ring current either remains constant or decreases due to pollutant adsorption on the Pt ring, necessitating cleaning of the Pt ring every few hours. If the ring isn't contaminated, why was there a need to replace the electrolyte in the experiment?

Response: In the rotating ring-disk electrode (RRDE) technique, a glassy carbon (GC) disk electrode with a Pt ring is employed as the working electrode. The catalyst is applied to the GC disk for the oxygen reduction reaction (ORR). To quantify the generated H₂O₂ at GC disk, the Pt ring is set at ≈ 1.2 V vs RHE, where the H₂O₂ oxidation on the ring electrode is detected. This results in a positive current at the Pt ring electrode, as H₂O₂ is produced at the disk (*Adv. Energy Mater.* **2018**, 8, 1801909).

During prolonged electrolytic reactions based on RRDE technology, there is invariably a portion of generated H₂O₂ that cannot be diffused to the ring electrode and enter the electrolyte without being oxidized. With the extension of time, the residual H₂O₂ concentration in the electrolyte continues to increase, and it will also be detected by the ring electrode, thereby increasing the ring current and 2-electron selectivity. This phenomenon is often observed in RRDE-based tests over extended durations (*Adv. Sci.* **2023**, 10, 2302446; *Adv. Energy Mater.* **2020**, 10, 2000789; *Adv. Mater.* **2023**, 35, 2209086). To ensure the long-term consistency of the ring current, we replace the electrolyte every 3 h to eliminate the impact of the accumulated H₂O₂ on the ring current. Additionally, as the reviewer notes, the Pt ring also needs to undergo electrochemical cleaning to eliminate any adsorbed contaminants. In response to the reviewer's concern, we have included corresponding explanations in the revised manuscript (highlighted by yellow).

(Page 10, Line 2-4) “During the continuous testing, the accumulated H₂O₂ might lead to a slight rise in the ring current and subsequently in selectivity. To maintain a consistent ring current, we replaced the electrolyte every 3 h and performed electrochemical cleaning of the Pt ring to remove any effects of the accumulated H₂O₂ during continuous operation.”

Comment 2-5:

Fifthly, in Figure 4i, Figure 4i showcases the disc current LSV of MHCS_x materials in 0.1 M PBS, presenting two sets of limiting currents with distinct slopes. The reviewer questions if this indicates the presence of varied reaction mechanisms at different potentials.

Response: We appreciate the reviewer's valuable comment. To address the reviewer's concerns, the ORR reaction mechanism was studied from the perspective of the H₂O₂ reduction reaction (H₂O₂RR). To investigate this, we carried out the H₂O₂ reduction reaction (H₂O₂RR) of MHCS_{0.5} in 0.1 M PBS containing 10 mM H₂O₂ under a N₂ atmosphere (*Angew. Chem. Int. Ed.* **2020**, *132*, 9256–9261). As shown in Supplementary Fig. 39, the small current densities indicate that MHCS_{0.5} is generally inert to H₂O₂ reduction. However, at more negative potentials, a slightly increased current density is observed. This finding confirms the reviewer's suggestion of different reaction mechanisms at varying potentials: near the onset potential, the 2e⁻ pathway is primarily followed, while at higher voltages, indirect 4e⁻ pathways occur slightly, leading to distinct slopes in the LSV (*J. Am. Chem. Soc.* **2021**, *143*, 20, 7819–7827; *Angew. Chem. Int. Ed.* **2015**, *54*, 6837-6841). Correspondingly, the variations in calculated H₂O₂ selectivity with potential (Fig. 4j) support this conclusion. We have incorporated the corresponding data and explanatory notes in the revised manuscript, highlighted by yellow.

Supplementary Fig. 39. LSV curves on MHCS_{0.5} for H₂O₂RR recorded in N₂-saturated 0.1 M PBS containing 10 mM H₂O₂.

(Page 10, Line 14-18) “Interestingly, the polarization curves exhibit two distinct sets of slopes at different potentials, suggesting variations in the ORR mechanism. Consistently, the H₂O₂RR test conducted on MHCS_{0.5} revealed a low H₂O₂RR current density, indicating its overall inertness to H₂O₂RR in a neutral electrolyte, with a slight increase observed at more negative potentials (Supplementary Fig. 39). This trend

in the H₂O₂RR current aligns with that of the ORR, highlighting the influence of the indirect 4e⁻ process at relatively high voltages on the 2e⁻ selectivity⁶¹.”

Comment 2-6:

Sixthly, based on the experimental design, variations in mass transport shouldn't impact the electrochemically active surface area, which could influence the active site concentration and limiting current. What caused this?

Response: We are grateful to the reviewer for this valuable comment. In our study of constructing mesoporous hollow nanoreactors, we manipulated the hollow ratio within MHCS_x. This manipulation inevitably resulted in variations in other morphological parameters, potentially influencing the contact with the electrolyte and consequently leading to different electrochemically active areas (*ACS Nano* **2018**, *12*, 10, 9635-9638). In accordance with the recommendations from this reviewer and Reviewer #3 (refer to **Comment 3-2**), we delved deeper into investigating the correlation between the surface area and the electrochemically active surface area of MHCS_x. As shown in **Supplementary Fig. 35b**, the electrochemically active area of MHCS_x did not exhibit a direct correlation with the BET specific surface area. However, upon employing the *t*-plot method (*J. Colloid Interf. Sci.*, **1996**, *21*, 405-414) to quantify external surface areas contributed by mesopores and macropores (excluding the contribution from micropores to the specific surface area), we observed a certain degree of positive correlation with the electrochemically active surface area (*Adv. Mater.* **2023**, *35*, 2301894). Considering the particle structures, it can be inferred that the increased electrochemically active surface area (*C_{ai}*) is closely associated with the enhanced exposed surface area facilitated by their mesoporous configuration, thereby promoting a more significant interaction with the electrolyte (*Angew. Chem. Int. Ed.*, **2020**, *59*, 5092–5101; *Nat. Synth.* **2022**, *1*, 658-667). Conversely, since micropores are not easily accessible to the electrolyte, their contribution to the electrochemically active area is limited (*ACS Sustainable Chem. Eng.* **2018**, *6*, 1, 311-317). To address the concerns raised by the reviewer, we have included a detailed discussion on the the electrochemically active surface area in the revised manuscript (**highlighted by yellow**).

Supplementary Fig. 35b, Total BET surface areas from Nitrogen adsorption-desorption isotherms, external surface areas calculated by the t -plot method⁵⁰ and double layer capacitance (C_{dl}) for MHCS_x. The external surface area refers to the BET surface area excluding the microporous area.

(Page 9, Line 14-19) “Interestingly, the C_{dl} values of MHCS_x do not align with their total BET surface area, but instead, exhibit a positive correlation with the external surface area excluding the microporous region (Supplementary Fig. 35b)⁶³. Considering the particle morphology of MHCS_x, it can be hypothesized that the internal surface exposure, facilitated by the mesoporous channels, enhances the interaction between the electrolyte and active sites, thereby promoting heightened electrocatalytic activity^{64, 65}. Prominently, the high C_{dl} value on MHCS_{0.5} signifies its structurally favorable diffusion, suggesting a swift supply of electrolyte ions and effective utilization of active sites.”

Minors:

Ag/AgCl is not a stable reference in alkaline solutions due to the formation of Ag₂O, which alters the reference potential.

Response: We agree with the reviewer’s professional comment. It is indeed true that the Ag/AgCl reference electrode is not typically employed in high concentrations of alkaline electrolyte (≥ 1 M KOH). However, the alkaline electrolyte used in this work was a 0.1 M KOH solution, which has a low concentration and therefore has little effect on the reference electrode. Additionally, as indicated in **Fig. R3**, our Ag/AgCl electrode was equipped with a salt bridge component to ensure their accuracy in RRDE testing. The utilization of salt bridges eradicates the liquid boundary potential and effectively minimizes mutual contamination between the solution of the study system and the solution of the reference electrode

system. In response to the reviewer's concern, we have included an explanatory note in the "Electrochemical measurements" section in the revised manuscript (highlighted by yellow).

Fig. R3. **a**, Ag/AgCl reference electrode equipped with a salt bridge. **b**, Ag/AgCl reference electrode equipped with a salt bridge in the RRDE system.

(Page 14, Line 4-6) "An Ag/AgCl electrodes was used as reference electrodes and equipped with a salt bridge to eliminate liquid boundary potentials and prevent cross-contamination between the solution of the study system and the reference electrode system."

Reviewer #3

General comment:

This study investigates the effect of geometric and porous structures of mesoporous hollow carbon nanospheres (MHCS) on two-electron oxygen reduction reaction ($2e^-$ ORR) activity for H_2O_2 electrosynthesis. The authors conducted fluid dynamics simulations to prove the advantages of mesoporous hollow structures, which could be attributed to facile O_2 supply as well as H_2O_2 emission. A set of MHCSs was prepared with carefully controlled structural properties, yet with similar surface oxygen functional groups, thereby enabling model catalyst study. It is demonstrated that $MHCS_{0.5}$ exhibits the highest H_2O_2 electrosynthesis activity owing to its optimal mesostructures which enables fast diffusion. The structure-dependent diffusion behavior was tried to be proved experimentally, where however, some issues should be addressed during the revision.

Overall, this reviewer would like to recommend the publication in Nature Communications, but after major revision. Detailed comments are listed below.

Response: We would like to express our sincere gratitude to reviewer #3 for the valuable comments and positive recommendation. In response to the reviewer's comments, we have incorporated additional experimental data and conducted thorough point-by-point revisions, accompanied by comprehensive and in-depth discussions.

Comment 3-1:

The density of mesopore is presumed to have a strong effect on the fluid behavior. This effect can be shown and briefly discussed in the revised manuscript.

Response: We sincerely appreciate the valuable suggestion from the reviewer. In response, we have conducted a more comprehensive analysis of the relationship between mesopore density and fluid behavior, considering both simulation and material perspectives. We constructed a modified model of $MHCS_{0.5}$, reducing the mesoporous density by a factor of 1 compared to the original model (Supplementary Fig. 12). Simulations were then conducted under identical conditions, with a focus on fluid velocities at specific locations within the mesoporous channel (-160, 0, 0 \rightarrow 160, 0, 0). Interestingly, our investigation revealed that the model with fewer mesopores maintained high flow velocities within the meso-channels for longer distances. However, we also observed an increased electrolyte flow from the surface of the sphere rather than through the mesopore channels, potentially reducing the frequency of contact between the mesopores and the electrolyte. Therefore, from a simulation perspective, a higher

mesopore density offers more mesoporous channels for the diffusion of the electrolyte fluid; conversely, an insufficient mesopore density may potentially decrease the utilization of active sites on the inner surface of mesopores.

In addition, as for a single MHN, it's evident that a small d/r ratio indicates a relatively long mesoporous channel and high mesoporous density. However, this configuration leads to continuous viscous resistance according to Newton's law of viscosity (*Nature* **1949**, *164*, 799-800), adversely affecting the flow rate (*Adv. Funct. Mater.* **2023**, 2213173). Conversely, a large d/r implies a relatively short mesoporous channel and a larger hollow structure with a lower mesoporous density. This architecture enables the fluid to swiftly enter the flow rate buffer zone of the hollow, albeit limiting fluid expansion.

According to the reviewer's suggestion, we have incorporated the relevant discussion into the revised manuscript (highlighted by yellow).

Supplementary Fig. 12. **a**, Schematic modeling of mesoporous hollow sphere ($d/r = 0.5$, $r = 150$ nm, $\phi = 20$ nm) with reduced mesopore density. **b**, Spatial distribution of fluid velocity in the hollow mesoporous sphere model with reduced mesopore density. **c**, Fluid velocity distribution across mesoporous hollow carbon sphere model with reduced mesopore density.

(Page 4, Line 14-21) “These findings indicate that the density of mesopores may influence the fluid behavior within the MHNs. For a single MHN, a small d/r ratio suggests a long mesoporous channel and high mesoporous density, resulting in continuous viscous resistance that affects the flow rate^{13, 46}. Conversely, a large d/r implies a short mesoporous channel and a larger hollow structure with lower mesoporous density, enabling rapid fluid entry into the hollow's flow rate buffer zone, though limiting fluid acceleration. Additionally, the modified MHN model, constructed with reduced mesopore density, demonstrates the electrolyte's ability to maintain a high flow rate over a long distance (Supplementary Fig. 12). However, the decrease in mesoporous channels would cause most of the electrolyte to flow from

the MHN's surface rather than through the mesopores, potentially reducing the utilization of the internal active sites.”

Comment 3-2:

The mismatch between the BET surface area and double layer capacitance trends of MHNs also suggests the effective diffusion of MHCS_{0.5} compared to other MHCSs because high double layer capacitance requires sufficiently fast supply of electrolyte ions. The authors can add some discussion on the relation between them.

Response: We sincerely appreciate the valuable input from the reviewer. We conducted further analysis of the surface area data of MHCS_x to better comprehend the electrochemically active area exhibited by MHCS_x. We agree with the reviewer that MHCS_{0.5} exhibits the highest double layer capacitance, implying efficient diffusion, and we have included a discussion on this aspect in the revised manuscript (highlighted by yellow).

Supplementary Fig. 35b, Total BET surface areas from Nitrogen adsorption-desorption isotherms, external surface areas calculated by the *t*-plot method⁵⁰ and double layer capacitance (C_{dl}) for MHCS_x. The external surface area refers to the BET surface area excluding the microporous area.

(Page 9, Line 14-19) “Interestingly, the C_{dl} values of MHCS_x do not align with their total BET surface area, but instead, exhibit a positive correlation with the external surface area excluding the microporous region (Supplementary Fig. 35b)⁶³. Considering the particle morphology of MHCS_x, it can be hypothesized that the internal surface exposure, facilitated by the mesoporous channels, enhances the interaction between the electrolyte and active sites, thereby promoting heightened electrocatalytic

activity^{64, 65}. Prominently, the high C_{dl} value on $MHCS_{0.5}$ signifies its structurally favorable diffusion, suggesting a swift supply of electrolyte ions and effective utilization of active sites.”

Comment 3-3:

It is wondered whether the H_2O_2 selectivity difference between $MHCS_{0.5}$ (the best) and $MHCS_{-0.5}$ (the worst) is demonstrated in flow cell experiments.

Response: We appreciate this valuable suggestion. To address this concern, we conducted additional tests on the performance of $MCS_{-0.5}$ for the electrochemical H_2O_2 production in a flow cell under alkaline (0.1 M KOH) and neutral electrolytes (0.1 M PBS) (Supplementary Fig. 43). Based on the test results, the electrochemical activity of $MCS_{-0.5}$ was found to be notably lower than that of $MHCS_{0.5}$, and its H_2O_2 selectivity was approximately 60% lower compared to $MHCS_{0.5}$ (~90%). Thus, the flow cell experiments also demonstrate a distinct difference in selectivity between $MHCS_{0.5}$ (the best) and $MCS_{-0.5}$ (the worst). We have included the relevant data and descriptions in the revised manuscript, which have been highlighted by yellow.

Supplementary Fig. 43. a, Polarization curves of $MCS_{-0.5}$ loaded CP electrode in flow cell. **b**, Electrolysis time-dependent current, H_2O_2 production, and faradaic efficiency of $MCS_{-0.5}$ under continuous O_2 purging in 0.1 M KOH. **c**, Electrolysis time-dependent current, H_2O_2 production, and faradaic efficiency of $MCS_{-0.5}$ under continuous O_2 purging in 0.1 M PBS.

(Page 10, Line 43-44; Page 11, Line 1-2) “In contrast, $MCS_{-0.5}$ exhibited restricted electrochemical activity, with currents of 51 mA at 0.1 M KOH and 25 mA at 0.1 M PBS at an applied potential of 0.1 V vs RHE

(Supplementary Fig. 43a). Additionally, the FE of MCS_{0.5} for H₂O₂ electro-production is approximately 60%, significantly lower than that of the MHCS_{0.5} electrode (Supplementary Fig. 43b, c).”

Comment 3-4:

The last experimental results where the MHCS_{0.5}-based electrode contained the largest amounts of OH⁻ ions (the side products of ORR in neutral solutions) among the samples are interpreted that MHCS_{0.5} catalyst had a poorer ability of exhausting products than the other samples, which is contradictory to the authors' claim. This reviewer guesses that this is just a result of faster electrocatalytic reaction with MHCS_{0.5} as a catalyst than the others. The authors should explain this and provide a time-current figure of the catalysts obtained in “Direct detection of the local pH changes on electrode” experiments.

Response: We agree with the reviewer's comments. To address the concern, we have included the time-current figure in the “Direct detection of the local pH changes on electrode” experiment (Supplementary Fig. 49a). Furthermore, we counted the current data obtained from the MHCS_x electrode and the corresponding OH⁻ concentration (Supplementary Fig. 49b). It is evident that the variation in electrocatalytic activity among the samples contributes to the distinct local OH⁻ concentrations, as suggested by the reviewer. Notably, the current density and OH⁻ concentration did not exhibit a strong linear correlation across the various samples (Supplementary Fig. 49c); notably, the MHCS_x electrode with a hollow structure demonstrated higher OH⁻ levels. Therefore, we attribute the localized increase in OH⁻ to the combined influence of electrochemical activity and the hollow structure. We have incorporated pertinent data and adjusted relevant explanations in the revised manuscript (highlighted by yellow).

Supplementary Fig. 49. a, Currents recorded in experiments for direct detection of the local pH changes on electrode. **b,** Detected current density and OH⁻ concentration at the MHCS_x electrode. **c,** Correlation between current density and OH⁻ concentration detected at MHCS_x electrodes.

(Page 11, Line 34-35) “Initially, as shown in Fig. 6e, the ORR process was performed for 10 minutes in a neutral electrolyte using a carbon sphere catalyst at 0.4 V vs. RHE, with the corresponding current density recorded in Supplementary Fig. 49a.”

(Page 11, Line 38-43) “This observation is associated with the high current density generated by the MHCS_{0.5} electrodes, leading to increased production of OH⁻ by-products. Additionally, it was observed that there is no linear relationship between the current densities and measured OH⁻ concentrations for each carbon sphere electrode (Supplementary Fig. 49b, c). Rather, the electrode with the hollow-structured carbon sphere exhibited a significantly higher OH⁻ concentration. Therefore, we attribute the localized OH⁻ accumulation on the MHCS_{0.5} electrode to the combined effects of its high electrochemical activity and favorable hollow geometric configuration.”

Comment 3-5:

*Similarly, the in-situ Raman results shown in Fig. 6h, where MHCS_{0.5} exhibited the highest signals for *O₂ and *OOH among the samples, were consequences of higher activity of MHCS_{0.5} than the other samples. Therefore, the authors should reinvestigate the results and the related sentences need tone-down.*

Response: We agree wholeheartedly with the reviewer’s comment that the *O₂ and *OOH signals of MHCS_{0.5} surpass those of all other samples, primarily attributed to its enhanced electrocatalytic activity. Guided by the valuable suggestion of the reviewer, the relevant sentences have been refined in the revised manuscript (highlighted by yellow).

(Page 12, Line 6-11) “This phenomenon can be inferred as follows: the enrichment of O₂ and localized pH elevation in the MHNs accelerate the reaction rate, facilitating the activation of O₂, stabilizing *O₂ intermediates, and inhibiting the protonation process of *OOH intermediates (*OOH + H⁺ + e⁻ → *O + H₂O) associated with the 4e⁻ pathway, thereby promoting H₂O₂ generation^{75, 79}. As illustrated in Fig. 6h, the comparison of the in-situ Raman spectra of all the samples revealed the most pronounced O-O stretching vibrations (ν*O₂ and ν*OOH) signals on the MHCS_{0.5}, indicating the accelerated reaction rate within the dominant conformation featuring favorable reaction microenvironments.”

(Page 13, Line 2-3) “Additionally, the accelerated reaction rate and favorable hollow configuration promote localized pH elevation, providing a suitable microenvironment for the 2e⁻ ORR.”

END OF RESPONSE

REVIEWERS' COMMENTS

Reviewer #1 (Remarks to the Author):

The authors have effectively addressed the concerns raised by the reviewers. I recommend this work to be published.

Reviewer #2 (Remarks to the Author):

The authors have appropriately addressed the concerns raised, providing comprehensive models, abundant EM images, electrochemical data, and detailed analyses. In my opinion, it is now suitable for publication.

Reviewer #3 (Remarks to the Author):

The authors responded satisfactorily to the issues raised by this reviewer, who now recommend the publication without further revision.